# How the extreme 2019-2020 Australian wildfires affected global circulation and adjustments

Fabian Senf[1], Bernd Heinold[1], Anne Kubin[1], Jason Müller[1], Roland Schrödner[1], and Ina Tegen[1]

[1]Leibniz Institute for Tropospheric Research, Leipzig, Germany.

**Correspondence:** Fabian Senf (senf@tropos.de)

**Abstract.** Wildfires are a significant source of absorbing aerosols in the atmosphere. Especially extreme fires, such as those during the 2019-2020 Australian wildfire season (Black Summer fires), can have considerable large-scale effects. In this context, the climate impact of extreme wildfires not only unfolds because of the emitted carbon dioxide, but also due to smoke aerosol released up to an altitude of 17 km. The overall aerosol effects depend on a variety of factors, such as the amount emitted, the injection height, and the composition of the burned material, and is therefore subject to considerable uncertainty. In the present study, we address the global impact caused by the exceptionally strong and high-reaching smoke emissions from the Australian wildfires using simulations with a global aerosol-climate model. We show that the absorption of solar radiation by the black carbon contained in the emitted smoke led to a shortwave radiative forcing of more than +5 $\mathrm{W\,m^{-2}}$ in the southern mid-latitudes of the lower stratosphere. Subsequent adjustment processes in the stratosphere slowed down the diabatically driven meridional circulation, thus redistributing the heating perturbation on a global scale. As a result of these stratospheric adjustments, a positive temperature perturbation developed in both hemispheres leading to additional longwave radiation emitted back to space. According to the model results, this adjustment occurred in the stratosphere within the first two months after the event. At the top of atmosphere (TOA), the net effective radiative forcing (ERF) averaged over the southern hemisphere was initially dominated by the instantaneous positive radiative forcing of about $+0.5\,\mathrm{W\,m^{-2}}$, for which the positive sign resulted mainly from the presence of clouds above the Southern Ocean. The longwave adjustments led to a compensation of the initially net positive TOA ERF, which is seen in the southern hemisphere, the tropics and the northern mid-latitudes. The simulated changes in the lower stratosphere also affected the upper troposphere through a thermodynamic downward coupling. Subsequently, increased temperatures were also obtained in the upper troposphere, causing a global decrease in relative humidity, cirrus amount, and the ice water path of about $0.2\,\%$. As a result, surface precipitation also decreased by a similar amount, which was accompanied by a weakening of the tropospheric circulation due to the given energetic constraints. In general, it appears that the radiative effects of smoke from single extreme wildfire events can lead to global impacts that affect the interplay of tropospheric and stratospheric budgets in complex ways. This emphasizes that future changes in extreme wildfires need to be included in projections of aerosol radiative forcing.

# 1 Introduction

Atmospheric aerosol particles from both anthropogenic and natural sources are important climate factors. The magnitudes of the different aerosol effects remain, however, highly uncertain (see Forster et al. (2021)). Aerosols influence the atmospheric radiation balance in different ways. The "direct radiative effect" refers to the change in the radiation budget of the Earth due to the scattering and absorption of incident solar radiation. An "indirect effect" of aerosol particles, on the other hand, results from the influence of aerosol particles as condensation nuclei for the formation of cloud droplets or for cloud ice and precipitation formation. While the overall average effect of aerosol particles on the radiation balance and surface temperatures is negative leading to reduction of surface temperatures (Forster et al., 2021; Thornhill et al., 2021), certain aerosol types, such as black carbon or soot particles, which originate from incomplete combustion processes and strongly absorb light in the spectral range of incident solar radiation, cause a positive radiative effect that counteracts the cooling caused by the non-absorbing aerosol types. This absorption of radiation results in heating rate changes within the aerosol-containing atmospheric layers. In turn, this heating causes rapid atmospheric adjustments to the instantaneous aerosol forcing, and has the potential to alter atmospheric dynamics and circulation. Such atmospheric adjustments are complex and include the effects of changes in heating rates on boundary layer stability or cloud cover. The various effects of the rapid adjustments or "semi-direct" absorbing aerosol effects have been summarized by Koch and Del Genio (2010). They can be studied with atmospheric circulation models on different scales that take into account the distribution of aerosol radiative properties. As an example, for a high resolution model experiment with grid resolutions of 312 and 625 m for Germany, Senf et al. (2021) found that the adjustment to heating by absorbing aerosol contributed positively to the effective radiative forcing with a similar magnitude as the instantaneous radiative forcing. Apart from changes in radiation budget forcing, such adjustments can also affect atmospheric dynamics and circulation. As an example for changes with global scale impacts, the effects of absorbing aerosol over the Indian subcontinent on the monsoon circulations have been analyzed in several model studies (Sanap and Pandithurai, 2015). In general, the atmospheric adjustment concept considers a combination of all atmospheric responses to forcings that are not mediated by the global-mean temperature (Sherwood et al., 2015).

Carbonaceous aerosol is produced in the industrial and agricultural regions mainly by burning fossil fuels or biomass (Bond et al., 2013). Sources comprise industrial and domestic emissions and traffic on the one hand, and vegetation fires including crop residue burning on the other hand. Climate change has been recognized as a new major driver for fire regimes worldwide (Jolly et al., 2015; Bowman et al., 2020). In addition to their immediate local disruptions of nature and lives and release of $CO_2$, wildfires produce carbonaceous particles that impact on the radiation balance (Sokolik et al., 2019). While anthropogenic aerosol from fossil fuel burning usually remains within the boundary layer, smoke from wildfires containing BC aerosol can be released into the atmosphere several kilometers high, and thus is subject to long-distance transport in the free troposphere (Huang et al., 2015). This is especially the case for high-intensity wildfires leading to deep convection due to the fire heat and eventually pyrocumulonimbus (pyroCb) formation (Fromm et al., 2010, 2019), potentially injecting a fraction of the smoke aerosols as high as the lower stratosphere. Such deep pyroconvection from extreme wildfires can have similar atmospheric

effects as aerosol from strong volcanic eruptions where the volcanic eruption plume reaches the stratosphere (Peterson et al., 2018).

Exceptionally strong vegetation fires occurred in Southeastern Australia in December and January 2019-2020 during the so-called Australia's Black Summer. During that fire season such pyroconvective smoke plumes were observed, injecting massive smoke amounts into the lower stratosphere (Kablick et al., 2020; Khaykin et al., 2020). Mainly between 29 December 2019 and 4 January 2020, several intense pyroCb events transported between 0.3 and 2 Tg of smoke particles (Peterson et al., 2021; Hirsch and Koren, 2021) up to 14–16 km height (Kablick et al., 2020; Boone et al., 2020). The resulting smoke layer extended across the southern mid and high latitudes, and could be detected in the stratosphere for as long as two years after the event (Ohneiser et al., 2022a). It had significant effects on the radiation budget. Its instantaneous positive radiative forcing in the southern hemisphere was estimated as high as $+0.5\,\mathrm{W\,m^{-2}}$ by Heinold et al. (2022). The actual radiative effect by the smoke aerosol would however be moderated by longwave adjustments in the stratosphere (Yu et al., 2021; Liu et al., 2022).

Beside the above mentioned value, a variety of other results for radiative forcing (RF) by Australian smoke can be found in the current literature. Reported values range from around $0.8\,\mathrm{W\,m^{-2}}$ (cloud case in Sellitto et al., 2022) to $-1.0\,\mathrm{W\,m^{-2}}$ (Hirsch and Koren, 2021). It has been argued by Sellitto et al. (2022) that the large spread can be attributed to uncertainties in the optical properties of the smoke, i.e., absorptivity and backscattered fraction. In our study, we aim to show in a complementary manner that the different assumptions used to estimate the radiative forcing, i.e., the consideration of cloud effects (Heinold et al., 2022), the contribution of resulting tropospheric and stratospheric adjustments to the forcing (Yu et al., 2021; Liu et al., 2022), and the spatial extent over which the averaged values were collected (applies e.g. to Hirsch and Koren, 2021; Liu et al., 2022), have significant influence on the estimated magnitude of the forcing value or even the sign of the radiative forcing. The dynamical mechanisms that led to the formation of a synoptic-scale anticyclonic smoke-induced vortex appear to be well studied (Khaykin et al., 2020; Kablick et al., 2020; Allen et al., 2020; Lestrelin et al., 2021; Doglioni et al., 2022), but it is unclear how the global dispersion of smoke impacted on the global circulation of the stratosphere and whether the perturbations in the lower stratosphere could have affected the upper troposphere. This chain of joint radiative and dynamical effects could represent an important mechanism for the downward coupling between the middle and lower atmosphere and help to better understand the climate effects due to extreme fire events.

This study extends the work of Heinold et al. (2022) by comprehensively quantifying the impact of fire aerosol on global circulation and adjustments using the same and additional experiments with the global aerosol-climate model ECHAM6.3-HAM2.3, which are described in Section 2. Emission characteristics are prescribed in our global modeling setup such that smoke is introduced into the lower stratosphere during the Australian pyroCb events (Section 2.2.1). To describe global smoke dispersion and understand emerging effects and couplings, both nudged simulations that relax the flow pattern towards an observation-based reanalysis (Section 2.2.2) and free-running ensemble simulations are conducted (Section 2.2.3). With the help of the simulation data, the regional and global radiative forcing is estimated and analyzed in Section 3.2 and compared with existing findings from the literature (Section 3.2.2). Furthermore, adjustments of the thermodynamic structure and the global circulation of the stratosphere (Section 3.3) and adjustments in the troposphere with respect to the hydrological cycle

(3.4) are analyzed. In Section 4 we discuss uncertainties, limitations and perspectives of our study and conclude in Section 5 with a summary and final remarks.

## 2 Data & Methods

### 2.1 Global Model ECHAM6.3-HAM2.3

ECHAM6.3-HAM2.3 is a global state-of-the-art aerosol-climate model (Tegen et al., 2019). In this study, we use the same model configuration as already detailed in Heinold et al. (2022). Global simulations were initialized on 1st October 2019 and integrated forward until 31st March 2020, i.e. six months ahead. The global model is set up and tuned for a horizontal resolution of approximately $1.875° \times 1.875°$ which results from the spectral truncation at T63. In the vertical, 47 levels are used from ground to 0.01 hPa. Model layers stretch in the vertical having around 70 m layer thickness close to ground and more than one kilometer in the lower stratosphere. This vertical resolution is quite coarse for resolving stratospheric processes, and a better vertical refinement of the middle atmosphere is envisioned for future studies.

The aerosol module HAM2.3 applies the aerosol microphysics model M7 (Vignati et al., 2004) for the prediction of several aerosol species, including sea salt, sulfate, mineral dust, organic carbon (OC) and black carbon (BC). The latter two are of special interest for the radiative impact of the investigated wildfire emissions. The aerosol microphysics scheme treats source processes (e.g. from fire emissions), transformations due to microphysics and atmospheric chemistry as well as loss processes, e.g. due to scavenging by clouds and precipitation, which influence the mass and number concentration of the aerosols in the model. The aerosol microphysics is coupled with a double–moment ice– and liquid–cloud microphysical scheme (Lohmann et al., 2007) to allow aerosol particles to interact with clouds. Aerosol particles also impact the atmospheric radiation fluxes. The particle optical properties are interactively calculated during model run time, taking into account their respective size and chemical composition. The aerosol particles are treated as internal mixtures, and optical aerosol properties such as extinction coefficient, single-scattering albedo, and asymmetry parameter are input from a Mie pre-calculated lookup table.

Atmospheric radiation is computed within ECHAM6.3 using the PSrad/RRTMG (Rapid Radiative Transfer Model for GCMs; Iacono et al., 2008; Pincus and Stevens, 2013) radiation package. 30 spectral bands are considered with 16 in the shortwave part of the spectrum, mainly influenced by solar radiation, and 14 in the longwave part of the spectrum, mainly influenced by terrestrial radiation. In addition to aerosol particles, also the optical properties of various gaseous atmospheric constituents and different hydrometeor types are also considered. For the analysis of the instantaneous direct aerosol radiative forcing, there is a double call of the radiative scheme to exclude atmospheric adjustments in the diagnosis.

### 2.2 Setup of Global Simulation Experiments

#### 2.2.1 Prescribed Wildfire Emissions

As in Heinold et al. (2022), the chosen global model setup applies prescribed wildfire emissions from the Global Fire Assimilation System (GFAS; Kaiser et al., 2012). GFAS assimilates observations of fire radiative power (FRP) based on MODIS

instruments aboard the polar orbiting NASA satellites Terra and Aqua to retrieve estimates of wildfire and biomass burning emissions on a daily basis. The GFAS emission data are input into the model as external data and mapped onto source descriptions of several aerosol species such as sulfate, dimethyl sulfide (DMS), OC and BC. Please note, however, that our modeled fire emissions neither represent a potential water vapor source nor perturbations in other gaseous compounds, such as ozone, carbon monoxide, etc.. This means that resulting phenomena such as the propagation of the water vapor anomaly, which has been observed by several space-born instruments (Schwartz et al., 2020), may be inadequately represented by our model data.

Wildfire aerosols are emitted mainly in the boundary layer and directly above it in the default model configuration. However, as argued in Heinold et al. (2022), the exceptional pyroCb activity of the extreme Australian fire event required to prescribe that most of the fire aerosol had to be injected at tropopause levels. Therefore, this study uses the approach that during the two phases (29–31 December 2019 and 4 January 2020) with strong pyroCb activity (as reported by Kablick et al. (2020)), the fire aerosol is directly introduced in the model layer above the tropopause with emitted aerosol masses of 0.6 and 0.2 Tg, respectively, according to GFAS analysis. Around 6 to 7 % of the total carbon load were attributed to highly absorbing black carbon. In the terminology of Heinold et al. (2022), the chosen emission setup is referred to as "TP+1" (i.e. full emission of fire aerosol in the model layer directly above the tropopause). Heinold et al. (2022) showed that similar aerosol effects can be found with other emission height specifications as long as most of the emitted absorbing aerosol is injected above the tropopause.

In order to make more robust statements about the resulting global fire effects, simulations were not only performed with the original GFAS emissions. As an extension, the fire emission strength during the pyroCb events was artificially increased by the factors 2, 3 and 5. Corresponding increases concern both organic carbon and black carbon emissions. The emission increases the impact on transport patterns and aerosol radiative forcings. The resulting additional simulation experiments were named according to the respective scaling factors with FIRE2, FIRE3, and FIRE5. Further experiments include FIRE1, in which the original GFAS emissions are used without scaling, and FIRE0 with zero Australian pyroCb emissions that is used as a reference to analyze the overall perturbations by the Australian wildfires.

It seems helpful to compare experiments with different scaling factors even if there is the risk that non-linearities influence or even distort the adjustments for higher fire strengths. A linear behavior becomes visible when the response of the system grows equally with the strength of the forcing. When rescaling is applied, in which fire-induced perturbations are divided by the fire scaling factor, e.g. by two for FIRE2 vs. FIRE0, all rescaled responses should be of similar size under the condition of small perturbations and linearity. If all rescaled experiments are subsequently averaged into a composite value, the lower noise of rescaled responses from the runs with the larger perturbations allow to increase statistical confidence.

### 2.2.2 Nudged Experiments

An important piece of evidence in our study originates from the so-called *nudged experiments*. For nudging, data from the ECMWF ERA5 reanalysis (Hersbach et al., 2020) were input as external data. Additional tendencies are introduced into the momentum equations such that the horizontal winds simulated by ECHAM are relaxed towards ERA5 winds. Surface pressure is also dragged towards ERA5 reanalyses. This strategy is applied to keep the model as close as possible to the observed meteorology. It is expected that more realistic transport pathways result in the nudged experiments especially because

prescribed fire emission agree well with simulated fire weather. Temperature fields were chosen not to be directly impacted by nudging in accordance to Zhang et al. (2014), such that no explicit temperature tendencies appear due to nudging. Nudging acts only as a small force on the dynamics. Important equilibria, such as geostrophy, are well preserved despite nudging. Dynamic adjustments of the system are possible even with nudging, but may be distorted by the effect of nudging. For this reason, it is not possible to isolate dynamic adjustments by comparing the nudged simulations with the free-running ensemble simulations presented later. However, an agreement of both simulation approaches indicates a robust response of the system.

The nudged experiments in our study are equivalent to the "replay" simulations mentioned in Doglioni et al. (2022) for the investigation of regional dynamical effects of absorbing smoke plumes. Similarly to Doglioni et al. (2022), freely running simulations were conducted for each experiment type, which will be discussed and contrasted with respect to their dynamical responses in the next subsection.

### 2.2.3 Ensemble Experiments

While the nudged simulations ensure a certain degree of realism of the resulting aerosol transport, they have their limitation for studying the effects of the injected wildfire aerosol on the atmospheric dynamics. For example, in Davis et al. (2022), nudging was shown to introduce biases into the representation of residual circulation and, consequently, tracer transport. These can only be evaluated from freely running *ensemble simulations*. However, this comes at the expense of probable discrepancies between the atmospheric state and the aerosol emission. As an example, the model might simulate rainy weather in South East Australia at the dates of the major wildfires and pyroCb events. Nonetheless, in order to obtain robust results of the dynamic response to the additional heating in the lower stratosphere using a freely running model, an ensemble approach was chosen.

Technically, all ensemble members share the same initial conditions as they were restarted from the nudged simulation on 1 Oct 2019. Tiny differences between the members are introduced by a variation (of order $10^{-4}\%$) of a parameter that governs the upward increase in stratospheric horizontal diffusion between adjacent layers, thus slightly perturbing mixing in the upper model layers. This perturbation does not alter the model climate, but introduces ensemble spread that fully develops within a few weeks after initialization.

For every fire scaling factor (0,1,2,3,5) an ensemble with 36 members was set up. Every member was run from October 2019 to March 2020 with the first two months being considered as spin-up time and, hence, excluded from the analyses.

All simulations, nudged and ensemble simulations share in common that prescribed sea surface temperatures and sea-ice concentrations were used at the lower boundary. These data were taken from the Atmospheric Model Intercomparison Project (AMIP) (Giorgetta et al., 2012). Long-lived greenhouse gases were defined following the Representative Concentration Pathway (RCP) 4.5 scenario.

## 2.3 Circulation Analysis

In the troposphere, the meridional overturning circulation is mainly fueled by strong meridional temperature gradients and is thus called thermally driven. In contrast, in the stratosphere the main driver that continuously induces a meridional transport of air is momentum deposition by breaking waves (Holton, 1997; Butchart, 2014). This is a key difference to the tropospheric

circulation. As it turns out, the conventional Eulerian mean of motion does not sufficiently take into account this additional eddy transport (Butchart, 2014; Matsuno, 1980), also known as Stokes drift (Dunkerton, 1978; McIntyre, 1980).

As a result of this under-representation of eddy effects, the circulation, calculated with the Eulerian mean, deviates strongly from the Lagrangian circulation deduced from tracer and mass transport in the stratosphere. A well established solution to this problem is the so-called Transformed Eulerian Mean (TEM), derived by Andrews and McIntyre (1976, 1978) as discussed in the review by Butchart (2014).

The TEM circulation can be understood as the part of the meridional Eulerian mean circulation that is not balanced by the Stokes drift and it is, thus, also referred to as the residual circulation (Butchart, 2014). Defined in this way, the calculated residual circulation is in much closer agreement with the Lagrangian mean paths of tracers compared to the conventional Eulerian mean whenever eddy activity is important. This makes the TEM a very well suited tool for the investigation of the circulation as well as possible circulation adjustments in the stratosphere. Following Peixoto and Oort (1992) and Grotjahn (1993), the residual wind speeds $\overline{v}^*$ and $\overline{\omega}^*$ are defined as

$$\overline{v}^* = \overline{v} - \partial_p\left(\overline{v'\theta'}\,\Gamma^{-1}\right), \tag{1}$$

$$\overline{\omega}^* = \overline{\omega} + \frac{\partial_\varphi\left(\cos\varphi\,\overline{v'\theta'}\,\Gamma^{-1}\right)}{a\cos\varphi} \tag{2}$$

where $\overline{v}$ and $\overline{\omega}$ are the zonally and monthly averaged meridional and vertical wind speeds (Eulerian sense), respectively. $p$ denotes atmospheric pressure, $\theta$ is the potential temperature, stability is related to $\Gamma = \partial_p\overline{\theta}$, $\varphi$ is latitude, $\partial_p$ and $\partial_\varphi$ are shortcuts for partial derivatives and $a$ the radius of the Earth.

Using these residual velocities one can now define a residual meridional mass stream function to quantify and display the stratospheric meridional overturning circulation e.g (Peixoto and Oort, 1992):

$$\overline{\psi}^* = 2\pi a\cos(\varphi)g^{-1}\int_p^{p_0} \overline{v}^*\,\partial p', \tag{3}$$

where $g$ is the acceleration due to gravity. The following section takes a closer look at the impact of the Australian wildfires on the atmospheric state including the circulation variables $\overline{\omega}^*$, $\overline{v}^*$ and $\overline{\psi}^*$.

## 3 Results

### 3.1 Modeled Distribution and Optical Properties of Australian Smoke

The Australian fire outbreak led to a significant input of black (BC) and organic carbon (OC) aerosol into the lower stratosphere. Estimates range from 0.3 to $2.1 \pm 1.1$ Tg (Khaykin et al., 2020; Peterson et al., 2021; Hirsch and Koren, 2021) which accommodates well the total emitted amount of 0.8 Tg used in our study. Once emitted from Southeast Australian fire hot spots, the smoke aerosol began to spread over the entire southern hemisphere reaching the southern tip of America after around ten days (Ohneiser et al., 2020). As a result, aerosol optical thickness (AOT) increased significantly in the otherwise relatively

pristine southern hemisphere (e.g., Hirsch and Koren, 2021). So, for example, the ground-based sun photometer observations of the AErosol RObotic NETwork (AERONET) showed a significant increase in AOT after a large peak around mid-January and persistent high levels thereafter (Heinold et al., 2022). In Punta Arenas, at the southern tip of South America, a monthly average of 0.1 was measured for the 550 nm AOT for January to March, an increase of more than a factor of 2 compared to 2019. Similar values of 0.08 - 0.12 and 0.04 - 0.065 on monthly average were also measured on Amsterdam Island and the Antarctic stations, respectively, during this period.

The ECHAM6.3-HAM2.3 simulations reproduce the evolution of the smoke plume with the pyroconvective injection heights prescribed. In the model, the stratospheric transport of the Australian wildfire smoke causes a monthly mean smoke AOT of up to 0.26 just downwind of the fire region in southeastern Australia in January 2020. Thereby, the plume dispersion is more confined and the smoke vortex is more pronounced in the nudged than in the free-running simulations. Persistently enhanced smoke AOT is simulated south of 30°S with values of 0.01 to 0.03 until March 2020 (Fig. 1). A significantly increased smoke AOT is also simulated over Antarctica due to the southward transport of the stratospheric smoke during the 3 month period considered. In agreement with the observations of, e.g., Khaykin et al. (2020) and Ohneiser et al. (2020), these model results indicate a considerable disturbance of the radiation balance in the southern hemisphere associated with the wildfire smoke burden.

For the 2019/2020 Australian wildfire plume, the model calculates a particle single scattering albedo (SSA) of 0.82–0.85 at 550 nm for the modeled ratio of BC to total carbon (BC=BC/OC) of 0.05–0.08 at the height of maximum smoke extinction ($\approx$16–24 km). The asymmetry parameter for the Australian smoke is about 0.6 at 550 nm, which is a typical value for wildfire aerosol (e.g., Reid et al., 2005). These parameters compare well the values derived from the inversions of the ground based multiwavelength lidar observations at Punta Arenas, Chile in January 2020 by (Ohneiser et al., 2020) and are also within the range of other aerosol models (e.g., Bellouin et al., 2020; Brown et al., 2021). For a detailed evaluation and discussion of the modeled smoke plume dispersal and optical properties we refer to the earlier study by Heinold et al. (2022).

## 3.2 Radiative Perturbations

### 3.2.1 Indications of Global Adjustments to Smoke-Radiation Interactions

The sudden increase in OC/BC burden due to the Australian fire outbreak leads to a significant rise in absorption of shortwave radiation in the middle atmosphere (MATM; in our case 200 hPa to TOA) that mainly localized over the Southern Pacific Ocean during January 2020 (see Fig. 2a and d) with large-scale average values of shortwave radiation flux convergence larger than 5 $\mathrm{W\,m}^{-2}$. Atmospheric mixing and transport lead to a more equal redistribution of stratospheric smoke aerosol in the subsequent months (Fig. 2b and c). Simulations with nudged meteorology indicate that significant amounts of smoke have been transported polewards with the implications that the bright Antarctic surface may have been darkened by stratospheric smoke when viewed from TOA. However, the poleward transport pathway is not as clearly visible in the ensemble simulations (Fig. 2d-f). This could be explained by the fact that, firstly, the fire emissions in the variety of ensemble realizations were not necessarily carried out during suitable fire weather conditions, but imposed at externally determined times independent of

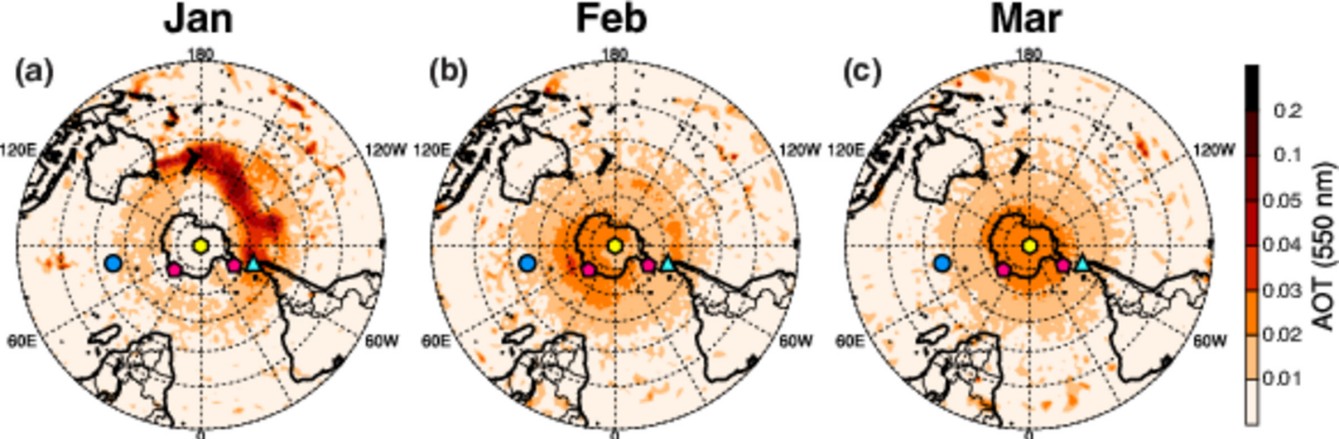

**Figure 1.** Maps of monthly mean smoke AOT for January to March 2020 calculated as the difference between the nudged ECHAM6.3-HAM2.3 simulation and a model run without the pyroconvective smoke injection in southeastern Australia. The location of the AERONET stations mentioned in the text is marked by colored symbols: Punta Arenas, Chile (53.14°S, 70.89°W), cyan triangle; Amsterdam Island (37.80°S, 77.57°E), light blue circles; Marambio (64.24°S, 56.63°W) and Vechernaya Hill (67.66°S, 46.16°E), magenta pentagons; South Pole (90.00°S), yellow hexagons.

large-scale conditions. Secondly, it is plausible that the coarse horizontal and vertical resolution of the climate model does not well represent the formation and movement of the smoke-induced vortex, as detailed in Khaykin et al. (2020), such that the flow field that is super-imposed by nudging is able to more realistically represent the smoke transport. An extended discussion of this aspect can be found in Sect. 4.

The average longwave forcing of the middle atmosphere (see Fig. 2g-l) appears to be weaker in amplitude than the shortwave forcing. It is mainly the nudged model runs that show a global negative longwave radiation response. The maximum amplitude was reached in February 2020 (Fig. 2h). The ensemble simulations confirm a negative longwave response in the southern hemisphere that appears to be weaker than the corresponding shortwave forcing. Towards February 2020, spatial areas increase in size in which nudged as well as ensemble simulations agree in the negative sign of the stratospheric longwave response,

also in the northern hemisphere. We interpret this signal as a fingerprint of an interhemispheric change in the stratosphere that can only be realized by changes in the global circulation pattern and thus need to be interpreted as global adjustments to stratospheric smoke-radiation interactions.

The situation becomes even more clear when larger smoke perturbations are considered (see Fig. 3). Increasingly larger radiative perturbations result from smoke emissions scaled with factors 1, 2, 3 and 5 (see Sect. 2.2.1). The time-average

radiative flux convergence in the middle atmosphere is dominated by shortwave absorption in the southern hemisphere between 60 and 20°S (Fig. 3d). The stratosphere responds with a negative longwave flux convergence that appears to be rather constant south of 45°S (Fig. 3e). However, we even find negative values of longwave forcing up to around 60°N. In total, a positive forcing still localizes at southern mid-latitudes, however, with a significantly smaller amplitude. This fact again indicates that

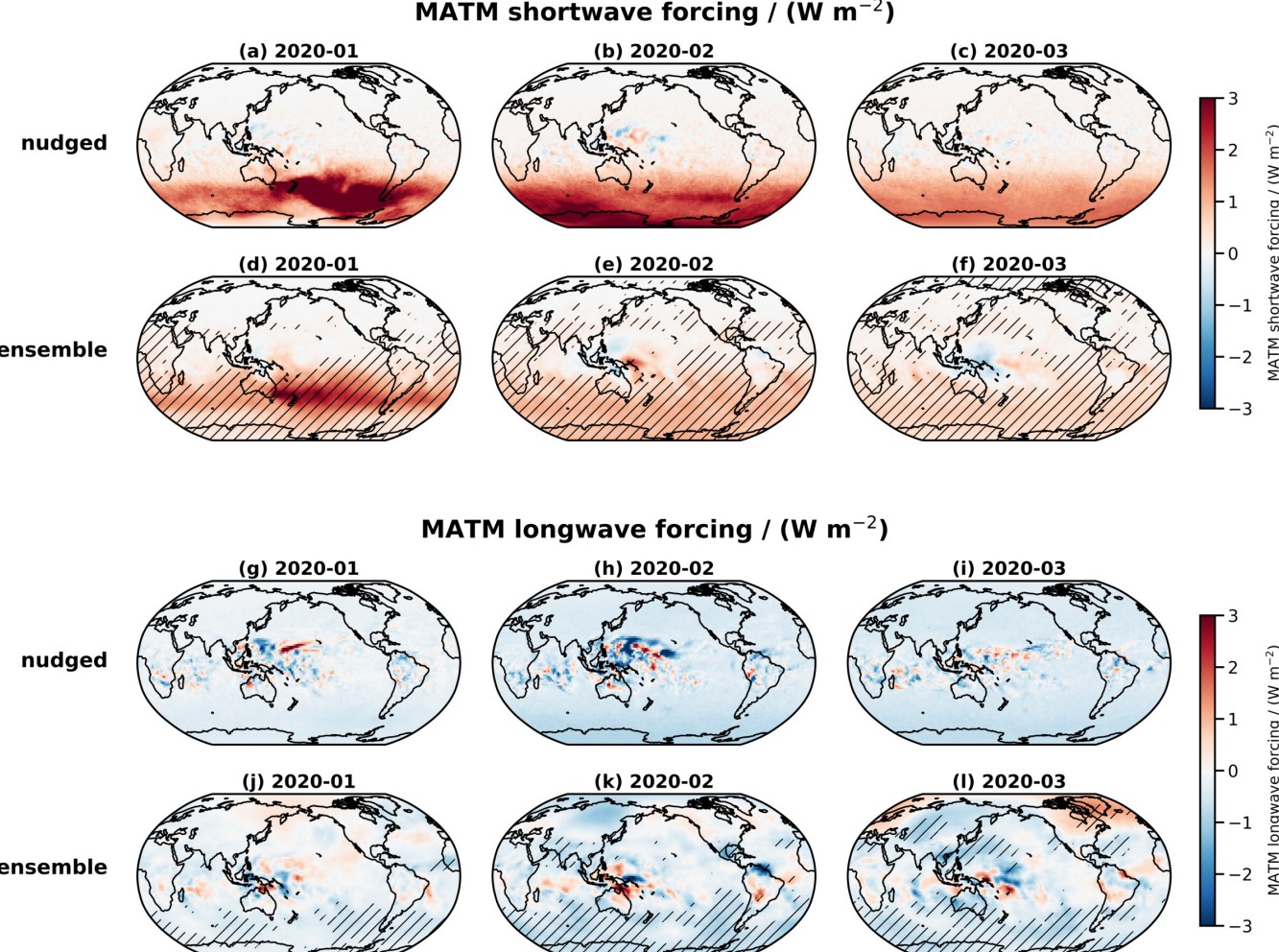

**Figure 2.** Convergence of monthly average radiative flux perturbations (difference between FIRE1 and FIRE0) between TOA and 200 hPa, abbreviated as middle atmosphere (MATM) forcing. The upper two rows from (a) to (f) present shortwave MATM forcing, the lower two rows from (g) to (l) are longwave MATM forcing. Time is increasing from left to right as indicated in the sub-panel titles, from January to March 2020. Average results from nudged (1st and 3rd row) and ensemble (2nd and 4th row) simulations are compared. Hatching represents areas in which the ensemble mean perturbations are different from zero at 95% confidence level.

the highly localized shortwave forcing is compensated to a considerable degree by a non-local (w.r.t. the smoke location) longwave forcing. In comparison to the middle atmosphere, forcings originating from the lower atmosphere are relatively small (Fig. 3g-j). The only component that contains perturbation signals is the longwave forcing. Due to the high uncertainties, it remains speculative whether a positive longwave forcing from the lower atmosphere compensates for some of the negative stratospheric forcing in the southern mid-latitudes. At the Earth's surface, dimming of solar radiation due to extinction by smoke is predominant in the southern mid-latitudes and has a maximum amplitude between 40 and 50°S. In total, the shortwave TOA

flux is positive (Fig. 3a) because the absorbed flux in the middle atmosphere more than offsets the reduction of shortwave fluxes at the surface. In relative terms, this reduction appears to be much smaller than reported elsewhere in literature (Hirsch and Koren, 2021; Yu et al., 2021; Liu et al., 2022), however, we will show in the next section that the presence of clouds is of major importance for the quantification of surface and TOA forcing of the Australian smoke event. The negative longwave TOA flux compensates the shortwave TOA forcing such that in total, the radiative forcing cannot be differentiated from zero

with confidence (Fig. 3b and c).

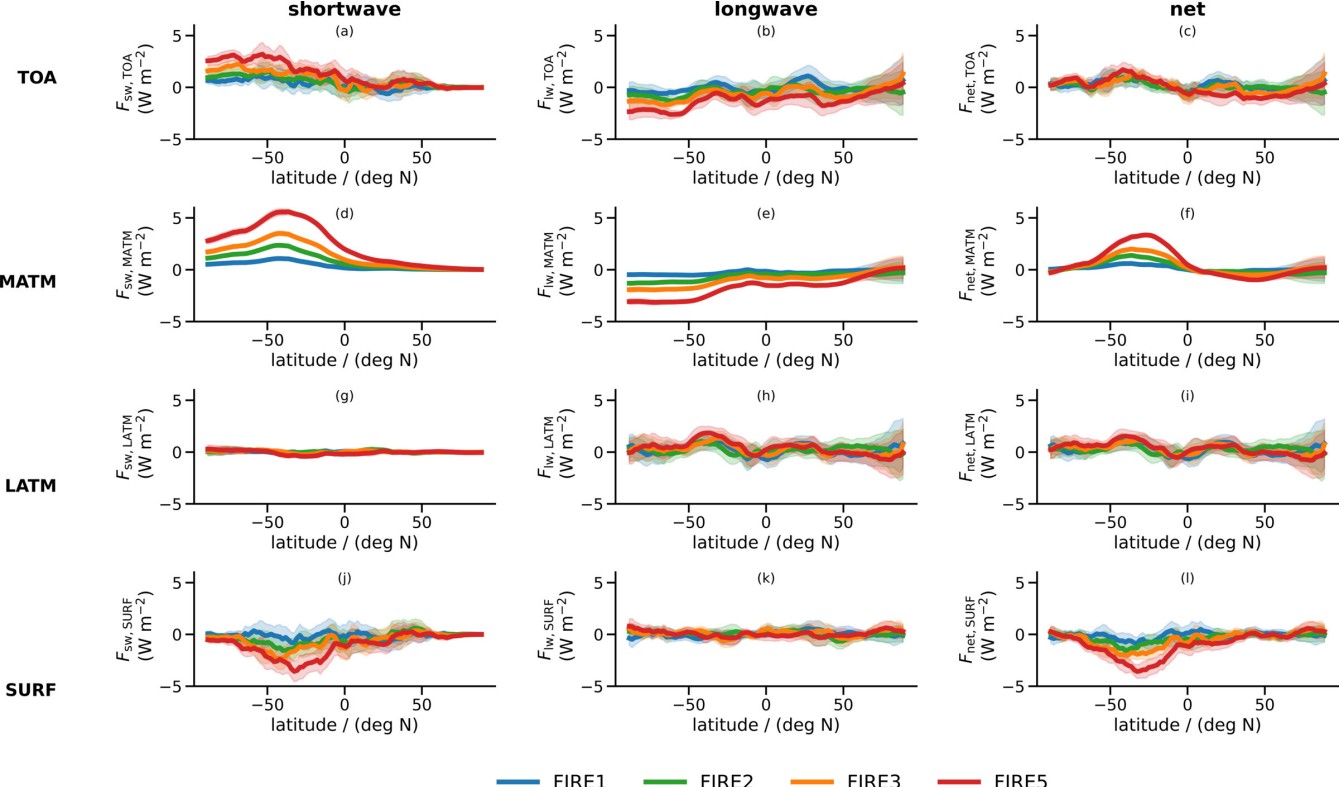

**Figure 3.** Ensemble and time zonal mean perturbations (JFM average) of radiative fluxes at TOA (1st row) and surface (4th row) and respective flux convergences between TOA and 200 hPa (2nd row), indicated as middle atmosphere (MATM) forcing, and between 200 hPa and surface (3rd row), indicated as lower atmosphere (LATM) forcing, as function of latitude. The colors represent deviations from the reference simulation (FIRE0) with different fire emission strength scaled by a factor of 1 (FIRE1: blue), 2 (FIRE2: green), 3 (FIRE3: orange) and 5 (FIRE5: red). The sum of shortwave (left column) and longwave (middle column) contributions gives the net contribution (right column). Thick lines represent the ensemble mean value, whereas light shadings show the 95% confidence interval.

### 3.2.2  Effective Radiative Forcing of Smoke and the Role of Clouds

In the following, we quantify and discuss the effective radiative forcing of the Australian smoke event based on a composite of all global simulation experiments where rescaled perturbations from different fire emission cases have been averaged. For this,

linearity of the atmospheric responses as assumed and perturbations divided by the respective fire scaling factor are averaged. The composite average across all experimental setups allows for a much more robust assessment of underlying effects, although non-linear responses may become more likely when fire emission strength is successively increased.

The effective radiative forcing and its components, instantaneous radiative forcing (IRF) and adjustments (Adj) are presented in Fig. 4a for all-sky conditions. In the southern hemisphere, the shortwave ERF is around $0.5\,\mathrm{W\,m^{-2}}$ and dominated by shortwave IRF. Adjustments become more and more important in the shortwave effects over time and contribute to around 30% of the total ERF in March 2020. In contrast, longwave ERF is dominated by adjustments and maximizes between February and March 2020. In comparison with the nudged simulations, the shortwave IRF appears to be smaller in the ensemble simulations due to less aerosol transport towards the southern high latitudes (see also Fig. 2a vs. d). In the temporal evolution of shortwave ERF, this mismatch seems to be compensated by stronger and more positive shortwave adjustments in the ensemble runs. In the tropics, ERF and its components are weak and uncertain in January 2020. In later months, longwave adjustments become more evident and contribute to a global negative longwave forcing. However, the net ERF in the tropics, for which substantial negative values were obtained in the nudged simulations, remains uncertain in the ensemble runs. In the northern hemisphere, the disagreement in ERF between nudged and ensemble simulation is largest which makes conclusions about the actually realized ERF considerably more uncertain. While the nudged simulations show similar longwave adjustments there as in the other regions, the ensemble simulations show a much weaker signal.

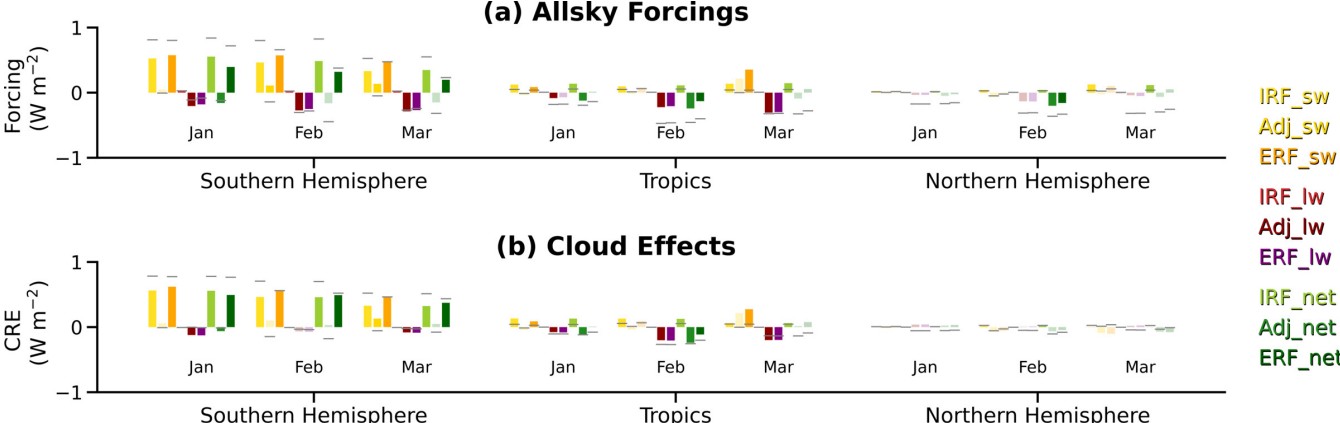

**Figure 4.** (a) All-sky forcings and (b) cloud effects averaged over specified geographical regions (outer interval on x-axis), over all months (inner interval on x-axis) and over all fire strength perturbations. The different components of the effective radiative forcing (ERF) are represented by differently colored bars. ERF is split into instantaneous radiative forcing (IRF) and adjustments (Adj) and also decomposed into shortwave and longwave components. Each bar represents the average of ensemble mean values for the four normalized fire perturbation experiments. Values for which the standard deviation among the four perturbation experiments is larger than the actual mean value are indicated by light colors. Data from the nudged experiments are shown as small gray horizontal lines for comparison and are also calculated as average over all normalized fire perturbations. Regions are split at $\varphi = 20°\mathrm{S}$ and $\varphi = 20°\mathrm{N}$ where $\varphi$ represents latitude.

Literature values of the net TOA radiative forcing from Australian wildfires are mostly negative (see Tab. 1). At a first glance, existing knowledge seems to be in conflict with our results. However, previous studies have mainly been limited to reporting either forcing for clear-sky situations, forcing that ignores atmospheric adjustments or forcing excluding the bright snow-covered surfaces in the polar regions. Previously, Heinold et al. (2022) had shown that the high total cloud cover over the Southern Ocean increases the planetary albedo so much that the dominance of scattering by smoke is lost. Thus, the original brightening of the dark ocean surface becomes a darkening of the bright, cloud-covered ocean and the negative shortwave forcing turns into a positive one (Bellouin et al., 2020). It is a known effect that increasing cloud cover also increases the instantaneous radiative forcing of absorbing aerosols that lie above the clouds (Chand et al., 2009). This effect is especially pronounced over the oceans. It becomes apparent from Fig. 4b in which the individual components of the clear-sky forcing have been subtracted from the respective all-sky forcings, in agreement with the definition of cloud-radiative effects (Stephens, 2005). The similarity between all-sky forcing and cloud effects is striking. It means that the clouds are the main driver of the positive forcings obtained in the southern hemisphere simply because of their presence, not regarding any cloud changes. In the tropics, the cloud effects significantly contribute to negative longwave adjustments because the cold cloud-top temperatures of the abundant cirrus increase the contrast of the warm perturbation in the lower stratosphere. As an intermediate conclusion, it can be stated that just the presence of clouds has a significant impact on ERF estimates and turns shortwave ERF into positive values in contrast to previously reported clear-sky values.

### 3.3 Adjustments in the Stratosphere

Shortwave heating by the extreme Australian smoke event triggers local and non-local temperature increases in the stratosphere. To illustrate the consequences for the energy balance, Fig. 5 shows the net heating rate, i.e. radiative heating due to convergence of shortwave and longwave fluxes, for the globally averaged lower stratosphere. A maximum between 0.03 and 0.04 $\mathrm{K\,(day)}^{-1}$ per fire scaling factor is reached after around one week after smoke injection. The nudged simulations seem to reach higher heating maxima for smaller fire strengths. In the following adjustments, a chain of processes acts to reduce the net stratospheric heating rate down to zero again. The turning point is reached equally fast for each fire strength after around nine weeks for the ensemble simulations or after six weeks for the nudged simulations. After the early peak in net heating, the decay follows approximately an exponential curve with decay constants of 14 and 21 days for the nudged and ensemble simulations, respectively. The nudged simulations appear to fall off faster as some of the smoke-induced shortwave perturbation leaves the lower stratosphere due to stronger self-lofting upward (see next paragraph) and as the longwave response also appears to be stronger.

As part of the stratospheric adjustments, a positive temperature perturbation emerges (see Fig. 6). Initially the temperature perturbation develops in the southern mid-latitudes in both the nudged and ensemble simulations, and subsequently spreads towards north and south in the following weeks. While the expansion of the positive temperature perturbation towards the south, i.e. in polar direction, can be mainly attributed to the transport of smoke aerosol and is therefore much more pronounced in the nudged runs, the expansion towards the tropics is related to the changes in the stratospheric circulation. The positive temperature perturbation reaches northern mid-latitudes after 10 to 14 days. Progression further north can also be seen in the

**Table 1.** Reported values of radiative forcing by Australian wildfire smoke are compared to our simulation results (first four rows). Our ERF estimates have been derived from the ensemble runs including the confidence interval with all values rescaled by the fire scaling factor. Literature values have been sorted in descending order.

| Reference | Radiative Forcing / $(\mathrm{W\,m^{-2}})$ | ERF | Clear-sky | Comments |
|---|---|---|---|---|
| FIRE1 | $0.2 \pm 0.2$ | * | | |
| FIRE2 | $0.06 \pm 0.11$ | * | | rescaled, globally averaged, JFM season |
| FIRE3 | $0.03 \pm 0.07$ | * | | |
| FIRE5 | $-0.01 \pm 0.04$ | * | | |
| Sellitto et al. (2022) | $0.8 \pm 0.2$ | | | simplified all-sky with const. surface reflectance at 0.5 |
| Heinold et al. (2022) | $0.44 \pm 0.07$ | | | global modeling, JFM season |
| Yu et al. (2021) | $-0.03$ | * | * | global modeling |
| Liu et al. (2022) | $-0.17 \pm 0.02$ | * | * | values averaged from $60°$N to $60°$S |
| Khaykin et al. (2020) | $-0.31 \pm 0.09$ | | * | area-weighted global-equivalent; instantaneous |
| Sellitto et al. (2022) | $-0.35 \pm 0.09$ | | * | area-weighted global-equivalent; instantaneous |
| Chang et al. (2021) | $-0.5 \pm 0.2$ | | * | rough estimate based on MODIS; instantaneous |
| Fasullo et al. (2021) | $-0.95 \pm 0.15$ | * | | likely no significant smoke emission into UTLS region |
| Hirsch and Koren (2021) | $-1.0 \pm 0.6$ | | * | observation-based; oceanic areas; 20°S-to-60°S belt |

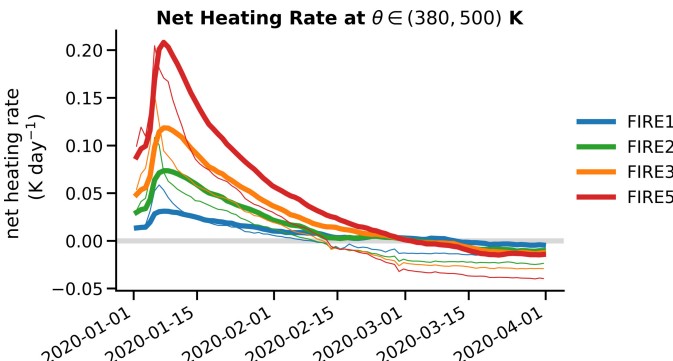

**Figure 5.** Net heating rate perturbation (i.e. the combination of shortwave and longwave radiation flux convergences) averaged globally for stratospheric altitudes between $380 < \theta < 500$ K as function of time. Colors indicate the different fire strength perturbations similar to Fig. 3. Thick solid lines are ensemble mean values and thin solid line represent averaged nudged data. The light gray horizontal line marks the zero line for orientation.

nudged runs. However, the considerable variability of the polar vortex in northern winter possibly hinders the attribution of any temperature effects in the ensemble simulations. A warmer northern hemisphere during 2020 was also identified in Rieger et al. (2021) based on observational data. Based on our analyses, we are wondering if the observed warming is not only a

coincidence but may be explained in part by smoke-induced circulation changes. Another difference between the nudged and ensemble runs is that the temperature perturbation is also more confined in the vertical direction in the ensemble simulations. This could indicate that the vertical self-lofting is more pronounced in the nudged runs, consistent with Fig. 2a-f in which
significantly larger shortwave forcings were found for the nudged runs.

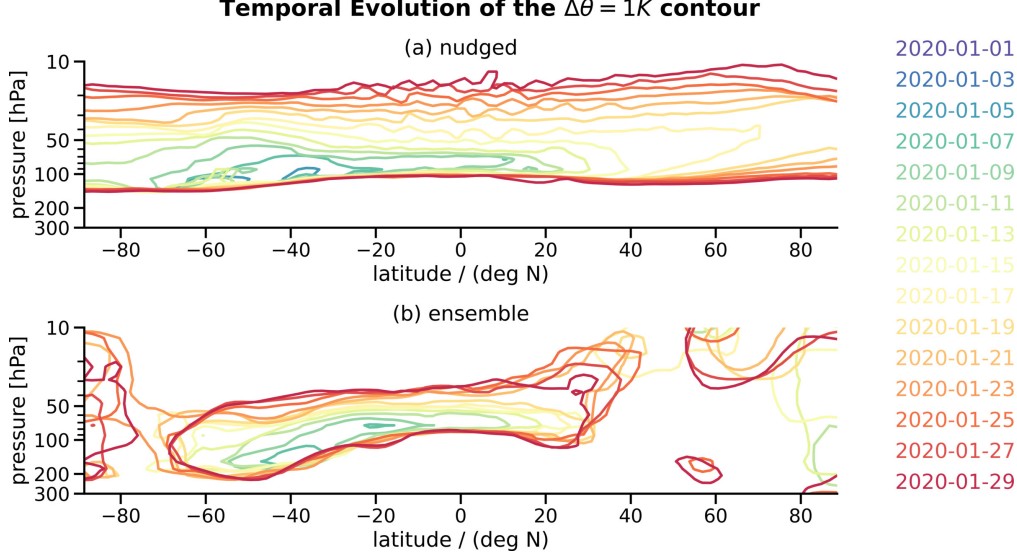

**Figure 6.** Temporal evolution of the potential temperature perturbation for the FIRE1 case with respect to the FIRE0 case. Shown is the 1-K contour of the average potential temperature perturbation for the (a) nudged simulations and (b) ensemble simulation that is color-coded by time (see legend). Each second day is presented beginning with 1 Jan 2020. Please note that the pressure at the vertical axis ranges between 300 and 10 hPa.

The stratospheric circulation changes become apparent from Fig. 7. In the southern tropics at around $10°$S, upward motion lofts air towards the tropopause region. Between 100 and 200 hPa, the mean flow divides into a northward and into a southward directed branch of the global atmospheric circulation. The northward directed branch reaches high into the stratosphere and is known as Brewer-Dobson circulation (Butchart, 2014). Nudged and ensemble simulation agree with confidence that due to
the effects of absorbing wildfire smoke, a positive anomaly of the residual streamfunction develops in southern mid-latitudes and in the tropics. This leads to a reduction of the strength of the poleward circulation in the southern hemisphere and to an increase of the circulation strength in the upper northern hemispheric branch. This increase propagates much further northward in the nudged simulations than in the ensemble simulations that are highly impacted by the variability of the winter vortex. Both, the nudged and the ensemble runs, also agree on a small weakening of the lower northward branch, but this finding is
highly uncertain as well.

Changes in the global circulation impact the energy budget of the stratosphere and especially how local diabatic cooling or heating by radiation is compensated by adiabatic heating or cooling from the global circulation. This aspect is visualized in Fig. 8. On average, lofting of air in the tropics leads to adiabatic cooling (dashed contours in Fig. 8) that counterbalances local

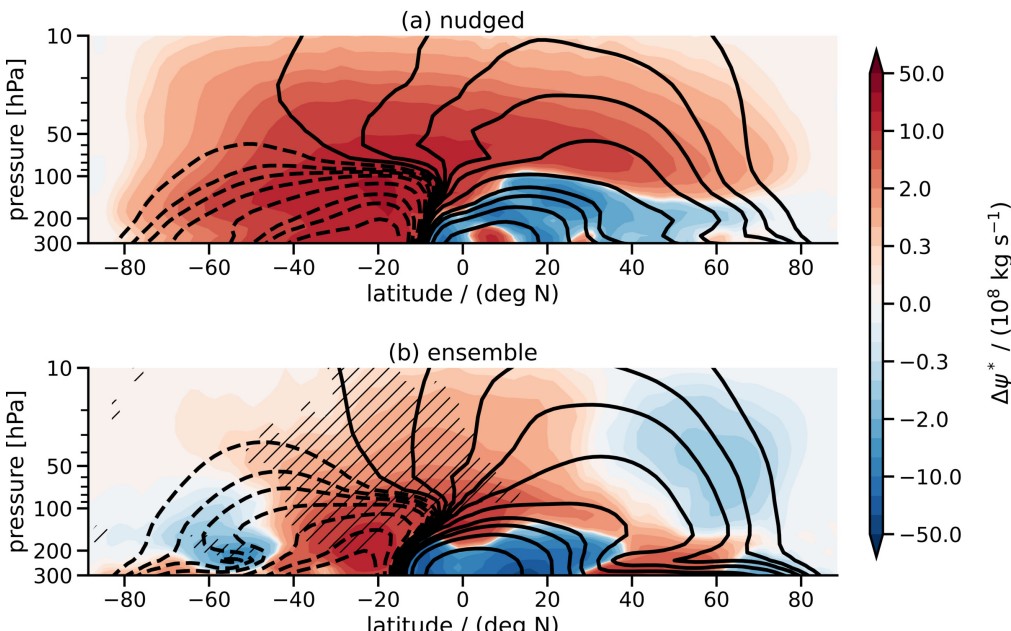

**Figure 7.** Average perturbations in residual circulation for (a) nudged simulations and (b) ensemble simulation as function of latitude and height. Contour lines indicate the state of the reference simulation FIRE0 with solid lines for positive values and dashed lines for negative values. Perturbations of the residual circulation in FIRE1 are presented in colored shading. Both, contours and shading have logarithmic spacing. Hatched regions have the same meaning as in Fig. 2.

diabatic heating by radiation and convection. Towards the poles, the average circulation acts to heat the stratosphere, thereby
compensating the dominant longwave cooling. This cooling is much more pronounced in the winter hemisphere which is why a stronger circulation is needed there for compensation. The circulation response (colored shading in Fig. 8) develops such that the excess of shortwave heating by smoke in the southern hemisphere is partly compensated due to adiabatic cooling. Moreover, the strengthening of the northward circulation causes additional adiabatic heating in the tropics and the northern mid-latitudes, thus redistributing the initial smoke-induced perturbation on a global scale. Such a temperature broadening mechanism by
circulation changes has been recently analyzed in detail for volcanic impacts on tropical dynamics (Brown et al., 2022). It is again highly uncertain how far this heating perturbation reaches northwards.

Due to adjustments, a new balance of the stratospheric state is established. Geostrophy is the strongest constraint for the zonally and monthly averaged dynamics. The positive temperature perturbation in the southern hemisphere causes the slope of the geopotential surfaces to increase toward the south pole, i.e. the perturbation of $\partial_\varphi \bar{\Phi}/a$, where $\varphi$ denote latitude and $\bar{\Phi}$ the
365 zonally and monthly averaged geopotential, is getting more negative. The resulting pressure-gradient force is nearly exactly balanced by the meridional component of the Coriolis force $-f\bar{u}$. Since the Coriolis parameter $f$ has negative values in the southern hemisphere, the perturbation of the mean zonal wind $\bar{u}$ also becomes negative and reaches values of a few meters per

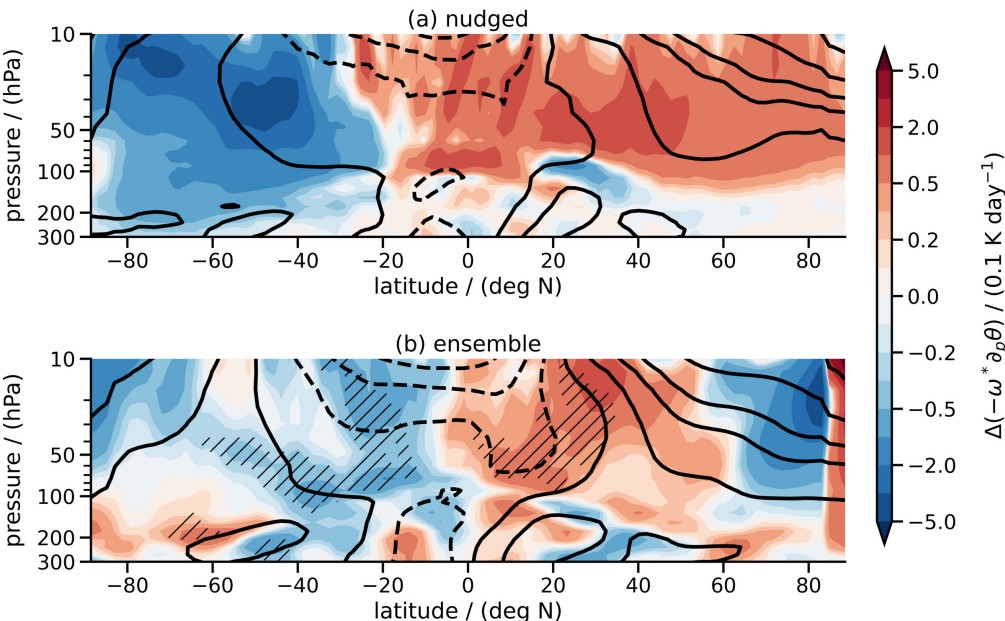

**Figure 8.** Similar to Fig. 7, but for the average perturbations in adiabatic heating due to circulation changes.

second in 100 hPa (not shown). The ageostrophic wind components $\overline{v}^*$ and $\overline{\omega}^*$ describe the zonal mean residual circulation (see Sect. 2.3). As shown in Fig. 9a, $\overline{v}^*$ is getting more positive in the southern hemisphere, thus the magnitude of the average

poleward motion is reduced and the southern-hemispheric branch of the mean circulation is slowed down (see also again Fig. 7). The sign of $\Delta\overline{v}^*$ is less clear in the tropics and in the northern hemisphere, and especially in the former region the response depends on the affected circulation branch. The significant positive correlations in Fig. 9a-c make clear that the adjustment of the mean meridional wind is accompanied by an adjustment of the large-scale waves measured by the so-called Eliassen-Palm flux defined after Edmon et al. (1980). The effect is only present in the freely running ensemble simulation, whereas nudging

destroys this important relation between the mean flow and the wave activity (not shown). The vertical component of the residual circulation is impacted differently by radiative heating depending on the region under consideration (Fig. 9d-f). In the southern hemisphere, the additional heating by shortwave smoke forcing induces increased upward motion (given by negative $\Delta\overline{\omega}^*$) that is strongest in January 2020 before longwave cooling starts to balance shortwave heating in the later months (Fig. 9d). Thus, the response of the southern hemisphere is thermally direct. The circulation change is responsible for an adiabatic cooling

induced by $\approx |\Gamma|\Delta\overline{\omega}^*$ with $\Gamma = \partial_p\overline{\theta}$ that compensates the smoke perturbation. However, an indirect relationship between radiative heating and circulation adjustments is found in the tropics (Fig. 9e). Thus, circulation changes generate additional adiabatic heating, which further amplifies the radiation-induced increase in temperature. Circulation changes remain uncertain in the northern hemisphere (Fig. 9f) and appear to be unrelated to radiative perturbations.

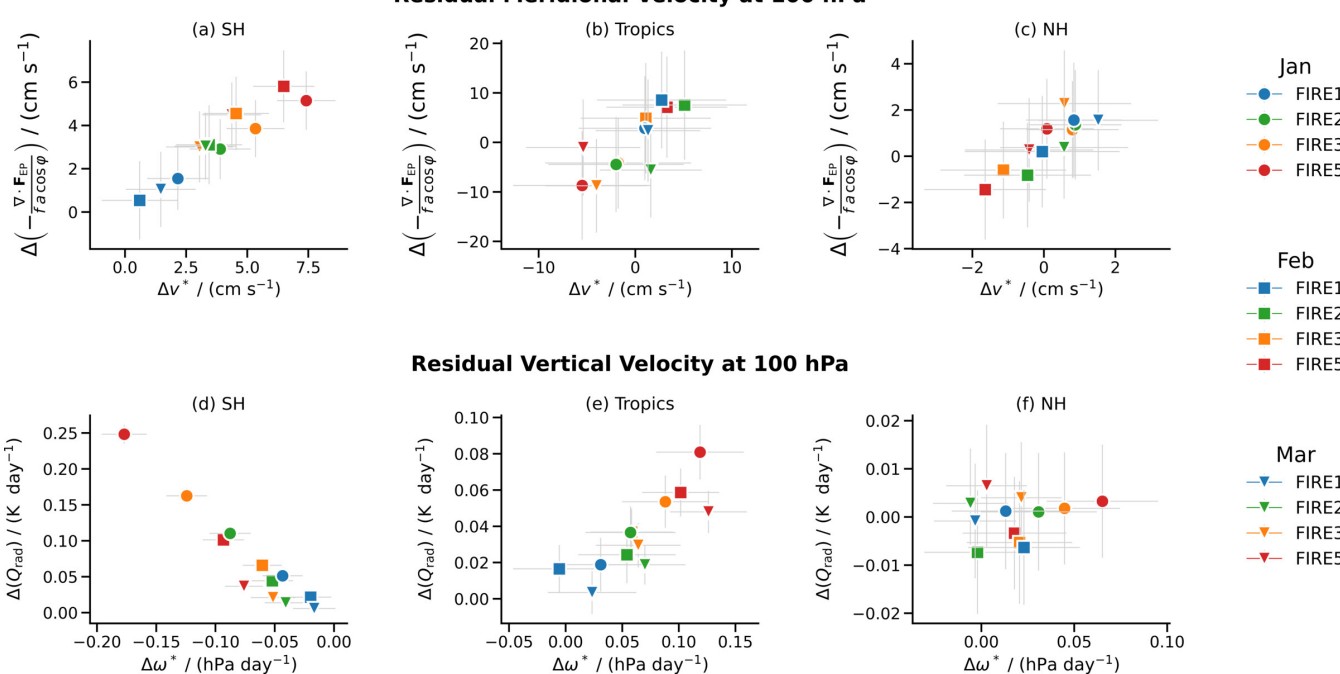

**Figure 9.** Relationship between components of the residual circulation and the forcing by large-scale waves (1st row) or the radiative heating (2nd row). Symbols mark monthly and regionally averaged anomalies with colors indicating fire strength from FIRE1 (blue) to FIRE5 (red) and symbol style for different months: January (circle), February (square) and March (triangle). Gray error bars indicate the 95% confidence interval. Regions are split by $\varphi = 20°\text{S}$ and $\varphi = 20°\text{N}$ where $\varphi$ represents latitude.

## 3.4 Adjustments in the Troposphere

In the following, tropospheric adjustments due to the Australian wildfire smoke are considered. Smoke-induced heating affects not only the lower stratosphere as described above, but also the upper troposphere at around 200 hPa. This is visualized by vertical profiles of radiative heating, potential temperature, relative humidity and cloud cover in Fig. 10. Again, the heating by smoke induces a positive temperature perturbation. This temperature anomaly not only travels upwards by self-lofting (as e.g. seen in Fig. 6), but also progresses downwards and subsequently enters the upper troposphere over the course of several

390 months as it can be clearly seen in Fig. 10b. The fact that the average temperature increases at 150 hPa and below despite the decreasing heating rates (see Fig. 10a) is an indication that effects due to changes in dynamics rather than sedimenting aerosol play an important role here. The warming of upper tropospheric layers leads to reduced relative humidity for which negative perturbations reach anomalous values of a few percent at 200 hPa (Fig. 10c). The reduction of supersaturated regions in the upper troposphere also reduces the coverage of the high clouds in the model, e.g. by about 1% at 200 hPa (Fig. 10d). Similar

aerosol effects on cirrus clouds were described in Samset and Myhre (2015) who applied perturbations by black carbon at

different atmospheric layers. Furthermore, Samset and Myhre (2015) pointed out that instantaneous forcing and adjustments differ the most when absorbing aerosol perturbations are introduced higher up in the atmosphere.

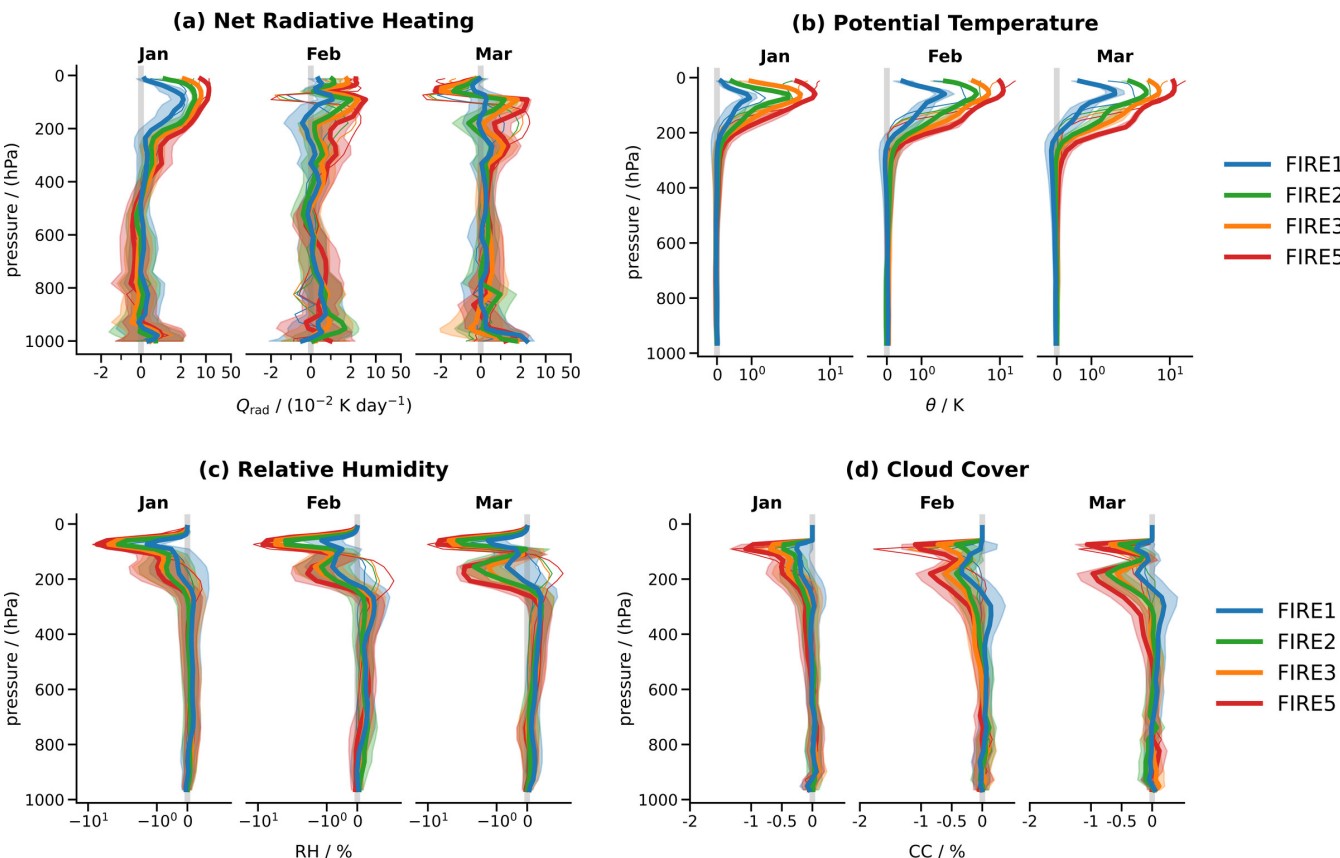

**Figure 10.** Globally-averaged vertical profiles of anomalies of (a) net radiative heating, (b) potential temperature, (c) relative humidity and (d) cloud cover as function of pressure. Similar to Fig. 5, thick colored lines represent ensemble mean perturbations with respect to the FIRE0 case. Temporal evolution from January 2020 (leftmost sub-panels) to March 2020 (rightmost sub-panels) is shown. Humidity is shown relative to saturation over liquid water. Please note that the x-scale is non-linear to improve visibility.

The simulated reduction in cirrus clouds impacts longwave as well as shortwave radiation. Due to the smaller amount of cirrus, a smaller amount of terrestrial longwave radiation is trapped in the troposphere. Thus, more longwave radiation is able to escape to space leading to a negative forcing of around $-1\,\mathrm{W\,m^{-2}}$ for a cloud-cover reduction of 1% (shown for the tropics in Fig. 11a). The compensation between shortwave and longwave radiative perturbations happens in such a way that the net cloud radiative effects remain rather unchanged (Fig. 11b), in line with known neutral to slightly positive net effects of cirrus (e.g. Chen et al., 2000; Stephens, 2005; Senf et al., 2020). The reduction of ice-containing clouds appears to be accompanied by a reduction of surface precipitation. However, this aspect is subject to considerable uncertainty (the majority of data points in Fig. 11c shows a negative precipitation anomaly, but a clear sorting with fire strength is missing). Over land, mixed-phase

and ice-phase cloud microphysical processes play an extremely important role in precipitation formation (Mülmenstädt et al., 2015). It is therefore not implausible that a reduction in cirrus cover is partially accompanied by a reduction in precipitation. The precipitation behavior is linked to a dynamical response of the Earth system (see Fig. 11c). For $\Delta\overline{\omega}^*$ at mid-tropospheric levels of 500 hPa, a negative correlation with precipitation is obtained (-0.9). A larger $\Delta\overline{\omega}^*$ means that the upward motion in the tropics is reduced. This dynamical response causes the mean circulation to contribute less to the adiabatic cooling of the tropical troposphere, producing a net dynamical warming of $Q_{\mathrm{dyn}} \approx |\Gamma|\Delta\overline{\omega}^*$. The greater this circulation-induced warming by $Q_{\mathrm{dyn}}$, the more atmospheric cooling $Q_{\mathrm{atm}}$ is compensated and the less precipitation must contribute to the warming of the atmospheric column (see Muller and O'Gorman, 2011). In general, the analysis of the energetic perspective of precipitation changes has proven to be very useful for understanding atmospheric adjustments to various forcing agents (Richardson et al., 2018), but a freely evolving model setup is needed because nudging destroys the link between precipitation and circulation changes. In the context of stratospheric aerosol intervention strategies, Haywood et al. (2022) could show that continuous injection of absorbing aerosols in the tropical lower stratosphere, does not only impact stratospheric temperatures, water vapor and ozone concentrations, but also affects the troposphere. Similar to our findings, absorbing aerosol in the stratosphere caused a downward influence on the troposphere leading to a reduction in surface precipitation.

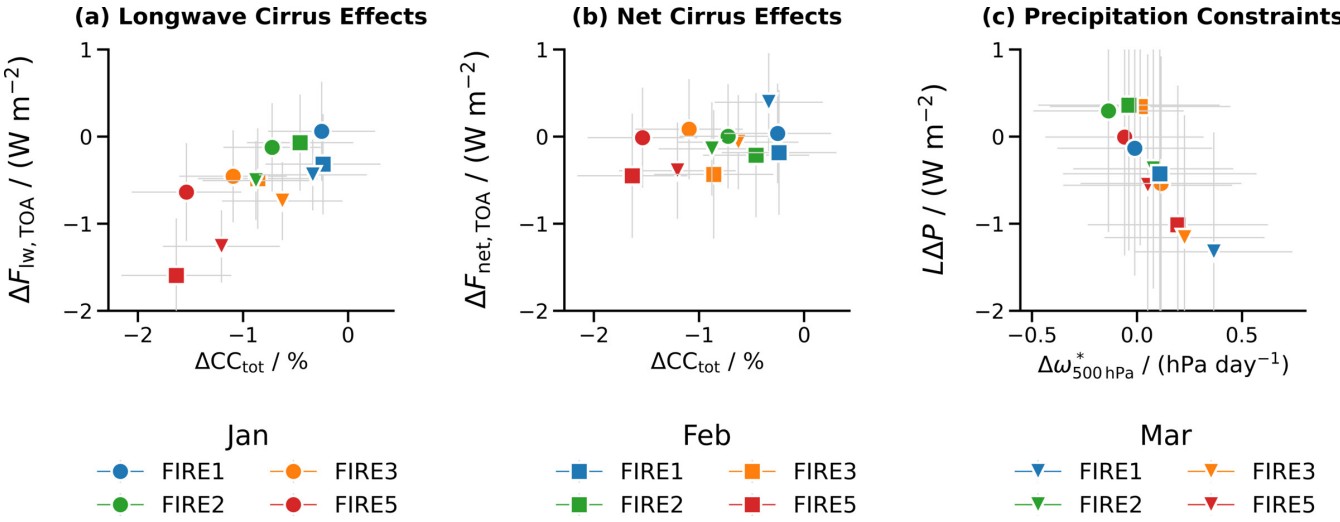

**Figure 11.** Cirrus effects on TOA radiation and energetic constraints for precipitation changes in the tropical belt set here to $\varphi \in (20°S, 20°N)$. Symbols and colors have the same meaning as in Fig. 9.

As seen above, not only cirrus cloud cover but the entire hydrological cycle is affected by the Australian fires, although the perturbations to evaporative and precipitation fluxes are quite uncertain (as indicated by the gray bars in Fig. 12a-c). Simulated total precipitation and evaporative fluxes are reduced by $-0.24 \pm 0.13\%$ and $-0.27 \pm 0.15\%$, respectively, for February and March 2020 on average where the interval provides bounds for confidence at 95 % level. For the different water phases, the clearest signal occurs for the ice water path which shows a systematic reduction of up to 1% in the FIRE5 run. Again, the plausible underlying cause is the temperature increase localized near the tropopause together with increased downwelling of

longwave radiation. These cause a slight perturbation of ice formation conditions such that less areas are supersaturated with respect to ice.

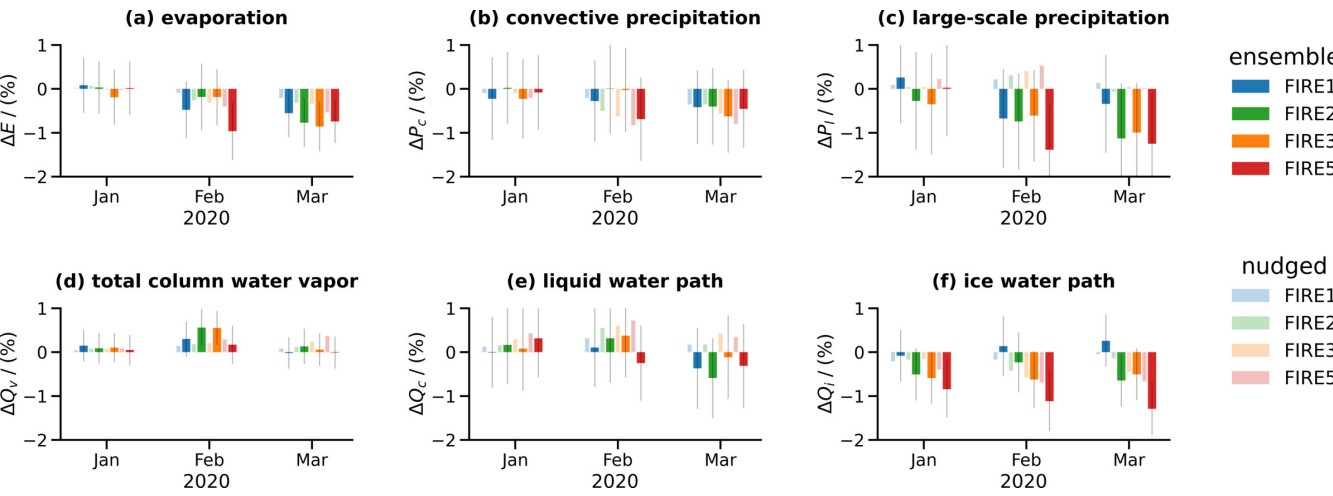

**Figure 12.** Globally and monthly average perturbations of (a) surface evaporation fluxes, (b) convective precipitation, (c) large-scale precipitation, (d) total column water vapor $Q_v$, (e) liquid water path $Q_c$, and (f) ice water path $Q_i$ as function of time. All perturbations are computed relative to the FIRE0 run as reference and presented in %. Colors from blue to red indicated perturbation resulting from fire perturbation runs FIRE1, FIRE2, FIRE3 and FIRE5. Bright colors indicate data from ensemble simulations, whereas light colors represent nudged data shown side-by-side for the respective runs.

## 4  Discussion

In the following, we will focus on the limitations of our study and discuss the consequences that follow. We deliberately chose
different modeling approaches to compile our results. On the one hand, we were able to model the dispersion of Australian smoke in a realistic way using nudged runs. On the other hand, free-running ensemble simulations allowed us to simulate the interaction of different thermodynamical and dynamical adjustments and also to estimate their resulting uncertainties. However, both approaches have their weaknesses and we aim to partially compensate them by using both approaches. One of the important weaknesses of the nudged runs is that they do not allow a free circulation response and thus reflect the dynamical
adjustments in a distorted way. This is especially the case for the relation between meridional residual wind and Eliassen-Palm flux divergence, which usually reflects a first order approximation to one component of the momentum balance (Edmon et al., 1980). In the nudged setup, the relaxation towards the observation-based mean winds and synoptic-scale wave structures dominates here. The free-running ensemble simulations eliminate this inconsistency. In those simulations, however, now the externally imposed fire emissions are not consistent with the meteorological situation. For example, precipitation may occur during
the extreme fire days in some ensemble realizations and not all runs have convectively unstable environments. Furthermore, we cannot answer the question whether the respective synoptic-scale smoke-induced perturbation phenomena are realistically

reproduced or whether the coarse model resolution prevents the formation and self-maintenance of the smoke-induced anti-cyclonic vortex. Doglioni et al. (2022) has shown that this phenomenon can also be described with coarser-resolution models which is promising. For our model data, however, this aspect remains a future research need.

As demonstrated by Heinold et al. (2022), the injection height of the wildfire smoke aerosol is critical for its atmospheric lifetime as well as its spatial distribution and, hence, for its radiative impact. Common injection heights for biomass burning are far below the tropopause leading to quick wet removal (within a few days) of the injected aerosol with only limited horizontal transport and impacts on atmospheric radiation. The strong upward lofting of smoke by exceptionally severe pyroconvection leads to injection of aerosol mass into the upper troposphere / lower stratosphere, where it has much longer life time. The

location above most precipitating clouds hinders wet and dry removal and a wider and even global dispersion becomes possible. Self-lofting due to heating by absorbed solar radiation (de Laat et al., 2012; Yu et al., 2019; Allen et al., 2020; Ohneiser et al., 2022b) further prolongs the presence of the smoke aerosol in the stratosphere and its impacts. Since we have no robust representation of pyroconvection and related transport processes in our model, the traditionally applied smoke plume injection was not able to reproduce the observed injection characteristics for the investigated wildfire event. For this reason, we had to

make the compromise that the pyroCb emission heights had to be prescribed in the model for the four most severe Australian wildfire days following Heinold et al. (2022). However, in order to be able to correctly simulate the climate impact of severe wildfires in climate projections, parameterizations are needed that reliably produce the correct frequency and characteristics of pyroCb events.

It must be mentioned that considerable uncertainties exist with respect to the microphysical and optical properties of wildfire

smoke. In the following, we reiterate the discussion detailed in Heinold et al. (2022) and again emphasize that even small variations in the content of highly-absorbing BC and in the morphology of smoke particles lead to significant changes in the global radiative forcing estimates (Sellitto et al., 2022). In our study, the ratio of BC to total carbon in Australian smoke is about 0.05-0.08, corresponding to a particle single scattering albedo (SSA) between 0.82-0.85 at 550 nm. In comparison to Bellouin et al. (2020) and Brown et al. (2021) these optical properties are similar to those used in other global aerosol-climate models.

However, Brown et al. (2021) demonstrated using aircraft data that the absorptivity of aerosol particles from biomass burning is overestimated by the models due to an inadequate description of the mixing state. However, these are generic results averaged over all possible biomass burning events that may not be easily extrapolated to extreme wildfires, since both the burned biofuel and the time for chemical processing are unique in these cases. This may also be the reason why Haarig et al. (2018) were able to derive a rather low SSA of 0.80 from multispectral lidar observations for the intense Canadian fires of 2017. Similar

low SSA values were also identified for the Australian extreme event (Ohneiser et al., 2022a). Furthermore, we also cannot exclude that secondary organic aerosol, which is not considered in our model, contributes significantly to the aerosol mass (especially in the later course) and leads to increased scattering by the aerosol mixture. Yet, the remarkable agreement of the various simulated smoke plume properties with observational data obtained by Heinold et al. (2022) further makes us confident that the absorption of solar radiation by the smoke aerosol is not massively overestimated in our model.

The Australian smoke impacted not only on stratospheric temperatures and circulation, but also on concentrations of strato-spheric water vapor and ozone. During the period from August to December 2020, a reduction in the total ozone column of 10

- 20 Dobson units was estimated for the southern hemisphere (Yu et al., 2021). First, changes in the Brewer-Dobson circulation (Diallo et al., 2022) may be responsible. Second, heterogeneous ozone chemistry at the surface of the smoke particles was identified as the potential cause of the record-breaking ozone hole over Antarctica in September-November 2020 (Ohneiser et al., 2020). Yu et al. (2021) estimated the contribution of the latter mechanism to be about 50% with respect to the total ozone reduction. However, since we do not describe stratospheric chemistry with our current model setup, and since ozone changes become more important in the second half of 2020, we only conducted simulation experiments up to and including March 2020. Therefore, we are only able to draw conclusions on the changes and underlying mechanisms during the first three months after the extreme Australian wildfire event. Furthermore, a substantially moister lower stratosphere was attributed to the effects of the Australian fires by Diallo et al. (2022), possibly due to both locally increased water vapor emissions from the extreme fires (Schwartz et al., 2020) and changes in water transport due to adjustments in global circulation.

In general, global adjustments can produce very diverse responses to perturbations in forcing constituents. The occurrence and impacts of severe wildfires are, however, highly dependent on details of the dynamical and thermodynamical state of the atmosphere as well as on details of the properties of the biosphere and the Earth's surface. These interconnections might make fire impacts less tractable as part of the global adjustment concept which might complicate the robust assessment of severe fire impacts in future climates.

## 5 Conclusions

Wildfires contribute to a significant amount to absorbing aerosol in the atmosphere (Bond et al., 2013; Sokolik et al., 2019). Absorption induces atmospheric adjustment processes that can have considerably diverse feedback on thermodynamic structure, cloud characteristics, and circulations (Koch and Del Genio, 2010; Bellouin et al., 2020). These adjustment processes can cause both positive and negative climate forcings and are still subject to large uncertainties (Smith et al., 2018).

The Australian wildfires were a particularly extreme fire event during the turn of the year between 2019 and 2020, which had a terrible impact on nature and the human population (Borchers Arriagada et al., 2020; Ward et al., 2020; Wintle et al., 2020). Due to the meteorological situation and extreme heat, strong pyroconvection clouds formed, efficiently transporting wildfire aerosol from the source of the fire to the upper troposphere and lower stratosphere (Kablick et al., 2020; Peterson et al., 2021). Once there, the fire aerosol had a significantly longer lifetime compared to smoke remaining in the lower troposphere, allowing it to disperse throughout the southern hemisphere (Ohneiser et al., 2022a). In this particular situation, localized causes had global and longer-term effects, which is why the Australian fires are compared to strong volcanic eruptions in the literature (Hirsch and Koren, 2021; Liu et al., 2022).

In contrast to aerosol of volcanic origin, the absorption of solar radiation by the black carbon contained in wildfire aerosol lead to relatively rapid adjustment processes in the stratosphere. That these caused a short-term increase in stratospheric temperature for the Australian event and thus compensate for the positive radiative flux convergence in the stratosphere is sufficiently documented by existing literature (Rieger et al., 2021; Stocker et al., 2021; Yu et al., 2021; Liu et al., 2022). However, the mechanisms leading to the adjustment, the resulting interactions between the lower stratosphere and the upper troposphere,

and the magnitude of the effective radiative forcing are not yet conclusively understood. Therefore, we would like to contribute our perspective on these aspects.

To shed light on the global effect of fire aerosol from the Australian wildfires, simulations were performed in our study using the global climate chemistry model ECHAM-HAM. Two separate modeling approaches were chosen and compared to reduce the uncertainties that arise. In nudged runs, where the simulated winds were relaxed to ERA5 data, the transport of smoke

was well reproduced. Additional free-running ensemble runs allowed us to examine adjustments in the global circulation and large-scale waves in more detail. Focusing on the smoke effects within the first three months after the Australian wildfire event, the following important results could be derived:

(i) The stratosphere responds to smoke-induced heating with a local and non-local warming leading to a temperature increase in the order of a few Kelvin. Compensating longwave cooling can offset a significant portion of the heating from locally

absorbed solar radiation.

(ii) Adjustments in the stratosphere lead to changes in the global circulation. In the southern hemisphere, less adiabatic heating by the residual circulation is required to compensate radiative cooling. Thus, the southern-hemispheric circulation branch is weakened. In contrast, the northern-hemispheric circulation branch is slightly strengthened such that the energy from the initially localized smoke-induced shortwave heating is redistributed to the tropics and the northern hemisphere.

(iii) In the troposphere, the amount of cirrus clouds is reduced by up to 1 % due to the warming effects of Australian smoke, which is caused by a reduction in relative humidity in the upper troposphere. The resulting tropospheric adjustments impact the hydrological cycle, with subsequently reduced amounts of ice water path, surface precipitation and evaporation. Due to energetic constraints, tropospheric circulation is also affected, with a link between precipitation changes and tropical upward motion leading to changes in adiabatic cooling in the tropics.

(iv) Clouds, just by their presence, have a significant impact on aerosol effective radiative forcing (ERF) estimates at the top of the atmosphere (TOA) and turn shortwave ERF to positive values in contrast to previously reported negative clear-sky values. However, atmospheric adjustments nearly balance the initially positive instantaneous TOA forcing by Australian smoke.

We showed that high reaching smoke plumes from pyroconvection can cause modifications in the stratospheric circulation.

The consequences of such large-scale circulation changes due to stratospheric smoke and potential impacts on the troposphere are not yet fully understood and further research is needed to disentangle the complex interaction and coupling mechanisms. As rapid climate change, affecting both the atmosphere and vegetation, increases the risk and intensity of wildfires (Jain et al., 2022), the representation of deep pyroconvective events and associated transport of absorbing aerosols into the stratosphere has to be improved in global climate models. Furthermore, the related consequences for the radiation budget, the changes in

the stratospheric circulation and the effects on the troposphere should be well understood in order to adequately consider these possible interactions in climate projections.

*Code availability.* The ECHAM-HAMMOZ code is maintained and made available to the scientific community under https://redmine.hammoz.ethz.ch (HAMMOZ community, 2022). The availability is regulated under the HAMMOZ Software Licence Agreement that can be downloaded from https://redmine.hammoz.ethz.ch/attachments/download/291/License_ECHAM-HAMMOZ_June2012.pdf (last access: 15 January 2023). The analysis source code has been made freely available to improve reproducibility of our results. The final plots for our paper were done with Jupyter Notebooks which are published at https://doi.org/10.5281/zenodo.7957666.

*Data availability.* Analysis data have been collected and can be assessed under https://doi.org/10.5281/zenodo.7568466.

*Author contributions.* FS, BH and IT conceived the idea and led the study. FS and AK performed the model development and ran the simulations. FS and JM focused on the analysis and interpretation of model results. FS, IT, RS, AK and JM wrote the paper with contributions from all the co-authors. All the authors participated in the revision and editing of the paper.

*Competing interests.* The contact author has declared that none of the authors has any competing interests.

*Acknowledgements.* The ECHAM-HAMMOZ model is developed by a consortium composed of ETH Zürich, Max Planck Institute for Meteorology, Forschungszentrum Jülich, the University of Oxford, the Finnish Meteorological Institute and the Leibniz Institute for Tropospheric Research (TROPOS) and managed by the Center for Climate Systems Modelling (C2SM) at ETH Zürich. We acknowledge computational support for conducting the experiments and storing the data at the Deutsches Klimarechenzentrum (DKRZ), within the compute projects bb1174, bb1262, bb1004 and bb1191. We further thank Kevin Ohneiser for his constructive feedback on an earlier manuscript.

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
