# Peer review of "How the extreme 2019-2020 Australian wildfires affected global circulation and adjustments"

_EGUsphere, 2023_

## Author Comment (AC1)

**"How the extreme 2019–2020 Australian wildfires affected global circulation and adjustments"**

**by Senf, F. et al.**

**Author's Response to the Editor and Reviewers of the manuscript**

Dear Editor and Reviewers,

We thank the Editor and the Reviewers for their comments on our manuscript. Based on your very valuable comments, the following general changes were performed.

- we expanded the description of the aerosol simulations
- we included a new subsection to discuss the simulated aerosol optical properties and how they compare with available observations
- we put more emphasize on the fact that our simulated precipitation response is still highly uncertain

In order to separate the reviewer's comments and the author's response we have used the following color coding and formatting. The reviewer's comments are printed in black, the authors responses are printed in blue and text parts taken from the updated manuscript are printed in green.

For the reviewer's convenience, we provide two additional files where changes between the original and updated manuscript are highlighted. First, the *diff.pdf* file contains the added and removed parts, the *diff-add.pdf* only contains the added content. The two files are compiled with degraded figure quality with *latexdiff* (unfortunately some artifacts remain) for fast difference checking.

Sincerely, on behalf of the authors,

Fabian Senf

senf@tropos.de

**Response to Reviewer #1:**

In this study, the authors use a climate model to simulate the large Australian wildfires of December 2019 and January 2020. They build on the study of Heinold et al. (2022), which looked at injection height and aerosol properties, to focus instead on rapid adjustments in the stratosphere and troposphere. They find that these adjustments are substantial and modulated by underlying clouds. Dynamical adjustments in the stratosphere are driven by different responses in the south and north branch of stratospheric circulation. In the troposphere, adjustments impact the water budget and may ultimately impact precipitation.

The paper is well written, and the description of the mechanisms of the response is generally convincing. Some aspects of the simulations and the discussion could however be clarified, as commented below. The revisions to address those comments should not require additional analyses, so should be minor revisions.

Main comments:

The aerosol simulations are not described. It is difficult to determine whether adjustments are local or not (e.g., line 227) when not knowing where aerosols are located in the simulations. A figure showing the extinction and absorption aerosol optical depth, or related variables, would be useful to follow the reasoning. In addition, it would be good to state the absorbing properties of BC and OC aerosols in the version of ECHAM6.3-HAM2.3 used in this study early on, rather than waiting for the discussion (lines 412-414).

> Thank you for this comment! We have firstly extended the description of the aerosol simulations in Section 2.1, and secondly, as suggested, we have added a results subsection in Section 3 showing the distribution of the simulated wildfire aerosol and discussing its optical properties.

The use of a combination of nudged and free-running simulations to isolate rapid adjustments should be presented more clearly in Section 2.2. My expectation was that dynamical adjustments are suppressed in nudged simulations, but not thermodynamical (temperature-driven) adjustments because temperature is not nudged. The comparison to free-running simulations therefore isolates dynamical adjustments. But the reasoning presented on lines 218-220 and 226-228 does not follow my expectations. Is it because dispersion is also different in the two sets of simulations? Overall, what does it say when both nudged and free-running simulations show similar responses?

> Nudging acts as an additional force that influences momentum balances and movements in the system. However, the nudging is weak enough so that important equilibria, e.g. geostrophy, are not significantly disturbed. The situation is different, for example, for ageostrophic wind components. There the nudging becomes visible already in first order which also means that nudging impacts the response of the global residual circulation.

Nudging changes the way the system adjusts and what new perturbed state it finds. However, it does not lead to the fact that no dynamic adjustments happen at all. It is too weak for that and a stronger nudging would possibly have negative effects on the whole model performance. The close coupling between thermodynamics and dynamics makes it difficult or impossible to separate the two aspects.

The following statement was added:

*"Nudging acts only as a small force on the dynamics. Important equilibria, such as geostrophy, are well preserved despite nudging. Dynamic adjustments of the system are possible even with nudging, but may be distorted by the effect of nudging. For this reason, it is not possible to isolate dynamic adjustments by comparing the nudged simulations with the free-running ensemble simulations presented later. However, an agreement of both simulation approaches indicates a robust response of the system."*

The simulated changes in stratospheric temperature and cirrus cloud cover are probably large enough to be observable. Are there observational studies that would support these findings?

We only partially agree with the statement above. Changes in stratospheric temperatures have been observed and attributed to the effects of the Australian wildfire by e.g. Rieger et al., (2021); Stocker et al., (2021) & Yu et al., (2021); see also Liu et al., (2022) for a synthesis. We mention these observational results in Sect. 3.3 & 5.

From our experience, however, we would argue that it is extremely difficult to identify such small changes in the cirrus cloud cover and very unlikely to causally link these changes to the effects of wildfires.

Other comments:

Line 41: The first instance of "radiative forcing" is ambiguous. Do you mean effective radiative forcing, or the radiative effect of the adjustments? From context I would say the latter, but that is not clear.

Thank you for this hint. We rephrased the sentence as following:

*"...absorbing aerosol contributed positively to the effective radiative forcing with a similar magnitude as the instantaneous radiative forcing..."*

Line 53: Is pyrocumulonimbus formation always happening when wildfire aerosols are injected high in the atmosphere?

From a cause-effect standpoint, we would turn the statement around to say that pyro-convection is able to lift wildfire aerosol to higher altitudes.

It is not clear for us, which part of the referenced statement led to confusion. We made it slightly weaker:

*"...potentially injecting a fraction of the smoke aerosols as high as the lower stratosphere."*

Line 76: I do not understand the use of "for" in this sentence. Isn't it the other way around, that radiative coupling between troposphere and stratosphere allows the chain of effect to happen?

In the case of Australian smoke, we see that dynamical and radiative effects are intertwined. As described later, Stratospheric circulation is affected by smoke heating. This likely causes also dynamical perturbations in the troposphere. Moreover, we also see that smoke partially heats the upper troposphere directly.

To make it clearer, we slightly rewrote the sentence:

*"... chain of joint radiative and dynamical effects could represent an important mechanism for the downward coupling..."*

Lines 135-136 and 241-242: I am not sure why the FIRE experiments need to be rescaled and averaged. Aren't the ensemble simulations enough to deal with statistical significance? And then check from the scaled FIRE experiments whether perturbations are indeed linear functions of the aerosol injection amounts?

Indeed, we have the problem that 1. ensemble runs in general have their weaknesses (see discussion) and 2. considerably more ensemble members are needed to improve the signal-to-noise ratio (x 100 for an order of magnitude better accuracy). The scaling applied can help increase confidence in the various simulation approaches by providing another third way in addition to statistical significance of the respective ensembles and in addition to the direct comparison of nudged and free runs.

In general, we think that the applied sensitivity experiments that scale the perturbation strength are very helpful to understand the system's responses. If the response R of the system is small than

$$R = S\lambda + \epsilon$$

where $S$ is a tangent linear model to describe the system's sensitivity, $\lambda$ is the perturbation and $\epsilon$ describes the inherent noise of system that does not go away due to limited average / ensemble size capabilities and also knowledge.

Rescaling by perturbation strength leads to

$$\hat{R} = S + \hat{\epsilon}$$

where the rescaled response $\hat{R} = R/\lambda$ is impacted by lower noise $\hat{\epsilon} = \epsilon/\lambda$. Thus,

combining simulations experiments with different forcing scalings can be more beneficial than just expanding the ensemble size by the same amount of model runs because the signal of the response may become clearer for the larger perturbation even even if there is a risk that non-linearities will influence the adjustments.

The following reformulation is proposed:

*"It seems helpful to compare experiments with different scaling factors even if there is the risk that non-linearities influence or even distort the adjustments for higher fire strengths. A linear behavior becomes visible when the response of the system grows equally with the strength of the forcing. When rescaling is applied, in which fire-induced perturbations are divided by the fire scaling factor, e.g. by two for FIRE2 vs. FIRE0, all rescaled responses should be of similar size under the condition of small perturbations and linearity. If all rescaled experiments are subsequently averaged into a composite value, the lower noise of rescaled responses from the runs with the larger perturbations allow increase statistical confidence."*

Lines 157-160: This is an unusual ensemble initialization technique. Does that risk making the ensemble too narrow by applying only a small perturbation?

Ensemble spread needs to build up with our perturbation method because we start with the same initial conditions. As shown in the Figure below, where we use standard deviation of global-mean OLR across all ensemble member as measure of spread, it takes not more than a month until the model spread is fully developed. This is because we run an atmosphere-only configuration with fixed SST and because of the fast error growth in atmospheric dynamics.

[Figure]

Lines 269-270: I understand how the mere presence of clouds modulate the SW forcing, but how does that work in terms of LW adjustments?

In the following we try to find simple words to describe the core of the mechanism for effects of clouds in the LW adjustments. However, we would like to emphasize that accurate radiative transfer calculations are necessary to quantify the effect exactly.

1. the absorbing aerosol heats a layer in the lower stratosphere which is located above clouds

2. LW radiation from below enters the heated layer, is partially absorbed and re-emitted there

3. thus a fraction of the LW radiation from below is replaced by radiation originating from the heated layer

4. effects of clouds:

   • clearsky radiation fluxes from below the layer would be rather high (warm emitter) whereas cloud-affected radiation fluxes would be lower (cold cloud top temperatures)

   • thus, the portion of radiation that is replaced by the heated layer (due to absorption and re-emission) is larger if clouds are present

A slight reformulation is proposed to make this aspect clearer:

*"In the tropics, the cloud effects significantly contribute to negative longwave adjustments because the cold cloud-top temperatures of the abundant cirrus increase the contrast of the warm perturbation in the lower stratosphere. "*

Lines 275-276: Just to confirm, that heating is directly due to absorption by the injected aerosols?

Figure 4 shows the pertubation of the net heating rate, i.e. the difference in net heating rate between e.g. FIRE1 and FIRE0. It includes shortwave and longwave radiative flux convergences, thus slightly more information than the absorption by injected aerosol goes in here. However, especially during the first weeks after injection, the net heating rate is dominated by smoke absorption. See also our answer below!

To make it clearer, the following re-formulation is supposed:

*"To illustrate the consequences for the energy balance, Fig. 5 shows the net heating rate,*

*i.e. radiative heating due to convergence of shortwave and longwave fluxes, for the globally averaged lower stratosphere. A maximum ..."*

Lines 279-281: And the heating decay is driven by the decrease in aerosol mass (or absorption)?

As stated above, Figure 4 shows the net heating rate which is the sum of the individual components of the radiation flux convergence. The split in shortwave and longwave heating rates is show below (material is also provided for download as part of the notebooks here: https://doi.org/10.5281/zenodo.7575809).

[Figure]

It becomes apparent that the decay of shortwave heating that is related to the decrease in aerosol mass is much slower.

Line 340: What range of pressures do you call the "upper troposphere" here?

We added:

*"...but also the upper troposphere at around 200~hPa."*

Lines 357-358: I do not understand what that sentence is saying, and what it implies for the results that were just presented.

To clarify the statement, the following re-formulation is proposed:

*"Over land, mixed-phase and ice-phase cloud microphysical processes play an extremely important role in precipitation formation (Muelmenstaedt et al., 2015). It is therefore not implausible that a reduction in cirrus cover is partially accompanied by a reduction in precipitation."*

Line 371: I would be good to specify which variability modes are referred to here.

The reference paper by Haywood and colleagues studied the impact of absorbing aerosol in the stratosphere and found reduction in surface precipitation as wellas changes in several climate variabilities modes such as the Quasi-Biennial Oscillation and North Atlantic Oscillation.

We like to connect to these findings with our study with the special emphasis on the reduction of precip. In order not to confuse the reader, we have deleted the second part of the statements.

Technical comments

Line 51: in several kilometers -> several kilometers

Corrected!

Line 59: smoke -> of smoke

Corrected!

Line 129: impact -> the impact on

Corrected!

Line 132: analysis -> analyse

Corrected!

---

## Author Comment (AC2)

**"How the extreme 2019–2020 Australian wildfires affected global circulation and adjustments"**

**by Senf, F. et al.**

**Author's Response to the Editor and Reviewers of the manuscript**

Dear Editor and Reviewers,

We thank the Editor and the Reviewers for their comments on our manuscript. Based on your very valuable comments, the following general changes were performed.

- we expanded the description of the aerosol simulations

- we included a new subsection to discuss the simulated aerosol optical properties and how they compare with available observations

- we put more emphasize on the fact that our simulated precipitation response is still highly uncertain

In order to separate the reviewer's comments and the author's response we have used the following color coding and formatting. The reviewer's comments are printed in black, the authors responses are printed in blue and text parts taken from the updated manuscript are printed in green.

For the reviewer's convenience, we provide two additional files where changes between the original and updated manuscript are highlighted. First, the *diff.pdf* file contains the added and removed parts, the *diff-add.pdf* only contains the added content. The two files are compiled with degraded figure quality with *latexdiff* (unfortunately some artifacts remain) for fast difference checking.

Sincerely, on behalf of the authors,

Fabian Senf

senf@tropos.de

**Response to Reviewer #2:**

The manuscript by Senf et al. investigates the radiative forcing and stratospheric circulation response to the extreme 2019-2020 Australian wildfires in nudged and free-running climate model simulations. The topic is of interest to ACP readership and the paper is clearly written. My main criticism is that the current manuscript lacks a comparison of aerosol optical properties between the different simulations and the observations. Without such a comparison, it is difficult for the reader to make an opinion on the realism of the different scenarios and to place the estimated forcings in the context of the literature. There are other limitations inherent in the study, and although most of them are briefly alluded to in the paper, the authors should consider expanding those points in the discussion.

For these reasons, I recommend that the paper be reconsidered after major revisions. My concerns are detailed below.

Major comments

1) It would be helpful to include a description of the evolution of the aerosol field in the different simulations and a comparison with the (already published) observations to assess how realistic the different runs are. CALIOP, SAGE and OMPS-LP are all suitable instruments and were used to characterize the Australian wildfire plume. This validation is necessary since the representation of the smoke plume in the large-scale model misses important processes which strongly affect dispersion, such as the confinement of the plume within vortices.

> Thank you for this remark!
>
> We assume that we have misleadingly not made clear that the present work builds directly on the studies of Heinold et al. (2022). The simulations used (in our nomenclature FIRE0 and FIRE1) are identical to the simulations of Heinold et al. (2022). This gives us the opportunity to rely on the evaluations conducted in Heinold et al. (2022) with observational data. There, simulation data were extensively compared with ground-based remote sensing (AERONET, Polly Lidar) and satellite-based remote sensing (AVHRR, CALIOP). It could be shown that our simulation approach reproduces the observations within the uncertainties.
>
> As a compromise, we have devoted an introductory results subsection 3.1 to simulated AOT and comparison with observations. However, we must emphasize that further detailed observation-simulation comparisons here do not fit the thematic focus of our paper.

2) An important amount of water vapor was injected together with the wildfire aerosols. A significant reduction in stratospheric ozone was also reported (e.g., Yu et al., 2021). As both water vapor and ozone are radiatively active gases in the stratosphere, they will impact the circulation response and stratospheric adjustment. Lines 432-434 of the manuscript, it is conceded that those effects are not well-represented in the simulations. However, I disagree

with the authors when they write that such perturbations "mainly occur in the second half of 2020" (lines 432-433). A number of papers (for instance, Kablick et al., 2020 ; Khaykin et al., 2020 ; Schwartz et al., 2020) documented that perturbations in composition were already present a few days after the injection. The authors could at least comment further on the impact that neglecting those perturbations may have on their results.

> To make clear that we do not emit additional water vapor together with the wildfires, we added the following explanations in Sect. 2.2.1:
>
> *"The GFAS emission data are input into the model as external data and mapped onto source descriptions of several aerosol species such as sulfate, dimethyl sulfide (DMS), OC and BC. Please note, however, that our modeled fire emissions do not represent a potential water vapor source and corresponding resulting effects such as propagation of a water vapor anomaly (Schwartz et al., 2020) may be inadequately represented by our model data."*
>
> In the discussion we wanted to express that we do not describe stratospheric chemistry and thus rather focus on adjustments on shorter time scales. To make that more clear, we reordered and rephrased to corresponding paragraph:
>
> *"... Yu et al. (2020) estimated the contribution of the latter mechanism to be about 50 % with respect to the total ozone reduction. However, since we do not describe stratospheric chemistry with our current model setup, and since ozone changes become more important in the second half of 2020, we only conducted simulation experiments up to and including March 2020. Therefore, we are only able to draw conclusions on the changes and underlying mechanisms during the first three months after the extreme Australian wildfire event. Furthermore, a substantially moister lower stratosphere was attributed to the effects of the Australian fires by Diallo et al. (2022), possibly due to both locally increased water vapor emissions from the extreme fires (Schwartz et al., 2020) and changes in water transport due to adjustments in global circulation."*

3) The treatment of the aerosols in the model should be presented in more detail in Sect. 2, in particular their interaction with chemistry and their radiative properties etc.

> Thank you for this comment! We have firstly extended the description of the aerosol simulations in Section 2.1, and secondly we have added a results subsection in Section 3 showing the distribution of the simulated wildfire aerosol and discussing its optical properties.

4) The precipitation response (Sect 3.3) does not seem very robust to me. Figure 10 and 11 show that, among the different simulations, it is not a monotonic function of the injected amount of black carbon aerosols. Is it really significant ? Furthermore, the mechanisms behind this response are not clearly explained. I would recommend either providing more information (and a mechanism) to support this hypothesis or shortening that point.

> Yes, indeed the precipitation response is very uncertain. When averaged over February and March, the following average results are obtained for the hydrological

variables:

[Figure]

Combining responses for the different fire strengths using rescaling as discussed in Sect. 2.2.1, a relative reduction of convective precipitation of -0.17 +- 0.2 % and a relative reduction of large-scale precipitation of -0.38 +- 0.32 % is found. I.e. the change in convective precip. is not significant, however the change in large-scale precip. is likely negative, however the magnitude is very uncertain. Both together have in fact a slightly reduced uncertainty: we find that total precipitation is reduced by -0.24 +- 0.13 %.

A reduction in precipitation is also in line with other studies by Samset and Myhre (2015) and Haywood et al. (2022), hence it is at least plausible that such responses may have happened also due to Australian smoke effects.

As a suggestion for improvements, we have shortened the section somewhat and emphasized the uncertainty in the precipitation response in several places.

Other comments

l4: 'as high as the stratosphere': Could you be more specific and provide an altitude range ?

With reference to Ohneiser et al. (2020), we changed the sentence as follows:

*"...but also due to smoke aerosol released up to an altitude of 17 km."*

L 13: 'in the Southern hemisphere..' :'averaged over the Southern hemisphere..'

Corrected.

l 18-19: consider being more quantitative here

We added the numbers from the re-scaled results in the abstract:

*"... Subsequently, increased temperatures were also obtained in the upper troposphere,*

*causing a global decrease in relative humidity, cirrus amount, and the ice water path of about 0.2\,\%. As a result, surface precipitation also decreased by a similar amount, which was accompanied by a weakening of the tropospheric circulation due to the given energetic constraints. ..."*

l 73 : Actually none of the cited papers explain the dynamical mechanisms behind the formation and maintenance of smoke vortices. They just describe the phenomenon in reanalysis or model simulations. A more insightful paper in that respect may be Lestrelin et al (ACP, 2021), which could be cited, although it does not provide a fully satisfactory explanation from a dynamical point of view either.

We added the suggested reference, but like to point out that Allen et al. (2020) provides an in-depth analysis of the vortex dynamical characteristics.

However, we also agree that scientific understanding for such complex dynamical phenomena  must be constantly improved and thus changed the wording such as:

*"... mechanisms ... appear to be well studied"*

L 101-104: Could you include more detail (part of the table) ?

We added a few more sentences in the model description. Furthermore, the new section 3.1 also adds more details for describing the aerosol simulations.

L 121-125: Please recall the level of injection (the reader should go to Heinold et al. only for the details) and add a reference for the aerosol mass here. Also, the end of the sentence seems to be missing a verb.

We included the suggested additions.

L 133: I don't understand this sentence. Why can one not compare the different experiments when the response is not linear ? Would all the experiments be useful if the response were linear (would 2 not be enough)?

Yes, you are right! The considered sentence is indeed misleading.

We think that the applied sensitivity experiments that scale the perturbation strength are very helpful to understand the system's responses. What we actually meant was that the response scales with the perturbation strength and this may help attribute certain response pathways to the perturbation. If the response R of the system is small than

$$R = S\lambda + \epsilon$$

where $S$ is a tangent linear model to describe the system's sensitivity, $\lambda$ is the perturbation and $\epsilon$ describes the inherent noise of system that does not go away due to limited average / ensemble size capabilities and also knowledge.

Rescaling by perturbation strength leads to

$$\hat{R} = S + \hat{\epsilon}$$

where the rescaled response $\hat{R} = R/\lambda$ is impacted by lower noise $\hat{\epsilon} = \epsilon/\lambda$. Thus, combining simulations experiments with different forcing scalings can be more beneficial than just expanding the ensemble size by the same amount of model runs because the signal of the response may become clearer for the larger perturbation even even if there is a risk that non-linearities will influence the adjustments.

The following reformulation is proposed:

*"It seems helpful to compare experiments with different scaling factors even if there is the risk that non-linearities influence or even distort the adjustments for higher fire strengths. A linear behavior becomes visible when the response of the system grows equally with the strength of the forcing. When rescaling is applied, in which fire-induced perturbations are divided by the fire scaling factor, e.g. by two for FIRE2 vs. FIRE0, all rescaled responses should be of similar size under the condition of small perturbations and linearity. If all rescaled experiments are subsequently averaged into a composite value, the lower noise of rescaled responses from the runs with the larger perturbations allow increase statistical confidence."*

L 143-144: I am skeptical that the nudging of wind only does not affect the energy budget, since wind is directly related to temperature through geostrophic/thermal wind balance as mentioned later in the paper. The Zhang et al reference might not be sufficient here, since those authors did not consider a large stratospheric aerosol injection but rather the background aerosol state.

We completely agree with your comment. What we actually liked to say was that the nudging tendencies do not explicitly appear as heat sources.

We rephrased:

*"...such that no explicit temperature tendencies appear due to nudging."*

L 157: 'order 10^-6' : '10^-4 %' would make clearer that it is the relative variation which is meant

Corrected.

l 174: I would remove 'very popular'

Corrected.

Figure 1: MATM could be defined in the main text as well as in the caption.

Following your suggestion, we defined MATM as well in the main text (first paragraph

of results section).

Line 276-277: "The nudged simulations seem to reach higher heating maxima for smaller fire strengths": do you have any idea why ?

This is a question that we cannot answer with definitive certainty and still requires further investigation by us. However, we believe that it may be important for the development of the nudged runs that dynamic structures from the observations were imposed onto the simulations in such a way that the smoke remain less diluted and therefore could be found in higher concentrations for a longer time. This would lead to higher heating maxima and also greater lofting rates as discussed together with Fig. 5.

L 280 : This longer decay time in the free running simulation seems at odds with Fig. 1 (which suggest a longer lasting SW perturbation for the nudged simulation) and the expectation that the plume is more diluted in free-running simulations. Again, it would helpful to have a comparison of the aerosol field between the two sets of runs.

Two aspect play a role here:

1. we show net heating i.e. a faster decay of net heating can be rather caused by a stronger longwave response and

2. we only consider a certain height range between 380 and 500 K in the lower stratosphere such that for stronger shortwave heating stronger self-lofting occurs and and the heating perturbation leaves the analyzed area in an upward direction.

[Figure]

The net heating rate which is the sum of the individual components of the radiation flux convergence. The split in shortwave and longwave heating rates is show below (material is also provided for download as part of the notebooks here: https://doi.org/10.5281/zenodo.7575809).

The following statement was added:

*"The nudged simulations appear to fall off faster as some of the smoke-induced shortwave perturbation leaves the lower stratosphere due to stronger self-lofting upward (see next paragraph) and as the longwave response also appears to be stronger."*

line 340: what is the lifetime of the black carbon aerosols in the upper troposphere in the model ? Do you confirm that they are refilled by sedimentation from above ?

Burden of stratospheric BC and resulting SW heating behave very similar with regard to the decay characteristics. We find e-folding times of around half a year for both. For illustration, the SW heating is shown as log-linear plot below. The dashed lines are for guiding the eye and represent an e-folding time of 4 months.

[Figure]

Moreover, our model data provide the indication that temperature perturbation in the UTLS moves slightly downward with time. However, we do not confirm that is effect is caused by sedimenting smoke particles. Indeed, the picture is more complex because the temperature increases despite the fact that SW heating due to absorbing aerosol is decreasing [see left panel below for quantities averaged over the SH at ~180 hPa; (circle, square, triangle) = (Jan; Feb, March) connected by a gray line]. We think that dynamic heating due to changes in large-scale circulation can not be neglected here.

[Figure]

**(a)** Δθ vs. SWH        **(b)** Δθ vs. diabatic heating

The following sentence was added:

*" The fact that the average temperature increases at 150 hPa and below despite the decreasing heating rates (see Fig. 10a) is an indication that effects due to changes in dynamics rather than sedimenting aerosol play an important role here. "*

Line 365: See major comment 4

Please see our response there.

Line 402: You might consider citing De Laat et al (2012) regarding self-lofting. It is one of the first papers to mention this effect.

Thank you for the hint! We included the mentioned reference.

Figure 10: Please reproduce the legend here, so that the reader does not have to go to a different figure to interpret this one

Done!

l 395: data : model ?

Changed accordingly!

l 457-460: this sentence might rather belong in the introduction

We agree with you, but also do not see that the closing remarks lose quality because of this sentence.

l 481: remove 'as' ?

We made the part even a bit shorter.

References

de Laat, A. T. J., Zweers, D. C. S., Boers, R., & Tuinder, O. N. E. (2012). A solar escalator: Observational evidence of the self-lifting of smoke and aerosols by absorption of solar radiation in the February 2009 Australian Black Saturday plume. Journal of Geophysical Research, 117, D04204. https://doi.org/10.1029/2011jd017016

Kablick, G., Allen, D. R., Fromm, M., and Nedoluha, G.: Australian PyroCb Smoke Generates Synoptic-Scale Stratospheric Anticyclones, Geophys. Res. Lett., https://doi.org/10.1029/2020GL088101, 2020.

Khaykin, S., Legras, B., Bucci, S., Sellitto, P., Isaksen, L., Tencé, F., Bekki, S., Bourassa, A., Rieger, L., Zawada, D., Jumelet, J., and Godin-Beekmann, S.: The 2019/20 Australian wildfires generated a persistent smoke-charged vortex rising up to 35 km altitude, Commun Earth Environ, 1, 22, https://doi.org/10.1038/s43247-020-00022-5, 2020.

Lestrelin, H., Legras, B., Podglajen, A., and Salihoglu, M.: Smoke-charged vortices in the stratosphere generated by wildfires and their behaviour in both hemispheres: comparing Australia 2020 to Canada 2017, Atmos. Chem. Phys., 21, 7113–7134, https://doi.org/10.5194/acp-21-7113-2021, 2021

Schwartz, M. J., Santee, M. L., Pumphrey, H. C., Manney, G. L., Lambert, A., Livesey, N. J., et al. (2020). Australian new year's pyroCb impact on stratospheric composition. Geophysical Research Letters, 47, e2020GL090831. https://doi.org/10.1029/2020GL090831

Yu, P., Davis, S. M., Toon, O. B., Portmann, R. W., Bardeen, C. G., Barnes, J. E., et al. (2021). Persistent stratospheric warming due to 2019–2020 Australian wildfire smoke. Geophysical Research Letters, 48, e2021GL092609. https://doi.org/10.1029/2021GL092609

---

## Editor Decision (ED1)

Editor comments ACP-2023-113

Senf et al., How the extreme 2019-2020 Australian wildfires affected global circulation and adjustments

Title: What do you mean with adjustments? I would suggest to rephrase the title.

P1, L18: "………..downward coupling mechanism in the model". Please rephrase. This sentence is misleading and sounds like that the model is affecting the atmosphere.

P1, L22: I am not sure if cycling is here the correct wording. Do you mean "exchange"?

P3, L61: To just write "adjustment" is not enough. You should write "according" adjustment and in general make clear that you mean with adjustment the adaption of the atmosphere to the new conditions.

P3, L87: Same here. Do you mean adjustments to the mode? Adjustments to the atmosphere?

P4, L94: add "as" -> as already detailed

P4, L09: input -> taken

P4, L114: What do you exactly mean with "optics" ? Optical properties? Please rephrase.

P5, L133: replace comma by semicolon? Better to rewrite the sentence so that it is better readable.

P5, L148: add "the" -> the so-called

P5, L149: "were input as external data" -> please rephrase. Better to just write "were used"?

P7, L189: as reviewed by -> better to write "as found" or "as discussed in the review by …"

P8, L224: computed -> simulated

P8, L225: I would suggest to be more clear and to write instead of "during the 3 months" "during the 3 month period considered".

P9, L251: nudged -> nudged model

P9, L282: I would suggest to write " In total". I think "In the net" is not correct English. It would be rather "In net".

P10, L269: singular or plural? Thus, either has "a" maximum amplitude or has maximum amplitudes.

P11, Figure 3 caption: add "forcing"

P12, L294: It is still not clear what is meant with "adjustment". Further, I would suggest to rephrase the sentence or move "there" behind "show".

P12, L295: At first glance -> At "a" first glance

P15, Figure 6 caption: Move "starts at 300 hPa" at the end of the sentence or write " pressure range between 300 and 10 hPa".

P18, L380: "Adjustment" to what?

P19, Figure 10 caption: plotted -> shown

P20,L412: low -> lower

P20, L412: make a listing instead of writing and twice? Thus, add comma after temperature and delete "and".

P20, L414: Downward impact of what? Do you mean with downward the circulation?

P20-21, L420-421: "downwelling of longwave radiation" sounds wrong.

P21, L427: Once again it is not clear what you mean with adjustments.

P22, L442: Impacts on what. Please clarify and rephrase sentence.

P24, L519: add "clouds" -> cirrus clouds

P24, L524: add "the" -> at the top

P24, L525: reported clear sky values -> where these negative? Please write clearly if negative or positive.

References: No consistent style used. Check ACP style and correct reference list accordingly.

---

## Author Response (AR2)

**"How the extreme 2019–2020 Australian wildfires affected global circulation and adjustments"**

**by Senf, F. et al.**

**Author's Response to the Editor and Reviewers of the manuscript**

Dear Editor and Reviewers,

we thank the Editor and the Reviewers for their comments on our manuscript. Based on your very valuable comments, we implemented changes for the 2nd and hopefully final revision.

In order to separate the reviewer's comments and the author's response we have used the following color coding and formatting. The reviewer's comments are printed in black, the authors responses are printed in blue and text parts taken from the updated manuscript are printed in green.

For the reviewer's convenience, we provide two tracked-changes versions of our manuscript: First, the 1st part of *diff-combined.pdf* file contains the added and removed parts. The second part of this PDF file only contains the added content. The PDF file is compiled with degraded figure quality with *latexdiff* (unfortunately some artifacts remain) for fast difference checking.

Sincerely, on behalf of the authors,

Fabian Senf

senf@tropos.de

**Response to Editor:**

from the letter:

There are some small issues left that should be taken care of before publication. Additionally, I would like to ask you to consider the corrections given in the attached pdf file. In general, I think you should carefully check your manuscript for the language. It is in many occasions not clearly written (feels always like something small but important is missing in the sentence) and the language itself is a bit too much slang (like discussing with your colleagues during coffee break or in a meeting, but not as it is supposed to be in a written document).

> We would like to emphasize that we are concerned that our scientific presentation is taken as casual language. We have therefore thoroughly read and reviewed the manuscript.

> However, it is also important to us that complex problems are presented as simple as possible. Figurative language can be helpful and ensure a better understanding of the context, especially for people who are not familiar with the subject or for non-native speakers. For us, simplicity is the highest premise and we are guided by the Einstein quote: *"Everything must be made as simple as possible. But not simpler."*

from the public justification document - Editor comments ACP-2023-113:

Title: What do you mean with adjustments? I would suggest to rephrase the title.

> With regard to the term "adjustments":

> Since AR5, the effects resulting from the aerosol-radiation interactions and the aerosol-cloud interactions are no longer divided into direct, semi-direct, and indirect effects. A change in nomenclature lead to the newly introduced re-partition into instantaneous effects and adjustments (see AR5, Fig. 7.3). This change in basic nomenclature in AR5/6 is certainly a call for the scientific community to adopt these terminologies accurately. A good introduction to the term "adjustments" is also provided in the paper by Sherwood et al, (2015, https://doi.org/10.1175/BAMS-D-13-00167.1).

> Honestly, therefore, we cannot see why the term "adjustment" should be ambiguous and do not want to choose a different designation in either the title or the text body.

> We added the following link at the end of our 1st introduction paragraph:

> *In general, the atmospheric adjustment concept considers a combination of all atmospheric responses to forcings that are not mediated by the global-mean temperature (Sherwood et al., 2015)*

P1, L18: "...........downward coupling mechanism in the model". Please rephrase. This sentence is misleading and sounds like that the model is affecting the atmosphere.

> Thank you for the hint! The misleading part was removed and the sentences was

shortened.

P1, L22: I am not sure if cycling is here the correct wording. Do you mean "exchange"?

We changed the word to "budgets".

P3, L61: To just write "adjustment" is not enough. You should write "according" adjustment and in general make clear that you mean with adjustment the adaption of the atmosphere to the new conditions.

We modified the part as follows:

*... resulting tropospheric and stratospheric adjustments ...*

P3, L87: Same here. Do you mean adjustments to the mode? Adjustments to the atmosphere?

The term "adjustments" is interpreted by us following Sherwood et al., (2015). We don't think this designation is unclear here.

P4, L94: add "as" -> as already detailed

done!

P4, L09: input -> taken

We use "input" as verb here. We don't think that our version is wrong.

P4, L114: What do you exactly mean with "optics" ? Optical properties? Please rephrase.

done!

P5, L133: replace comma by semicolon? Better to rewrite the sentence so that it is better readable.

We rephrased the sentence.

P5, L148: add "the" -> the so-called

done!

P5, L149: "were input as external data" -> please rephrase. Better to just write "were used"?

The verb "input" says exactly what we mean here, i.e. that we input the data as external data. We propose to keep it as is.

P7, L189: as reviewed by -> better to write "as found" or "as discussed in the review by ..."

done!

P8, L224: computed -> simulated

done!

P8, L225: I would suggest to be more clear and to write instead of "during the 3 months"

"during the 3 month period considered".

> done!

P9, L251: nudged -> nudged model+

> done!

P9, L282: I would suggest to write " In total". I think "In the net" is not correct English. It would be rather "In net".

> We choose "in total" here and also at a later occurrence.

P10, L269: singular or plural? Thus, either has "a" maximum amplitude or has maximum amplitudes.

> done!

P11, Figure 3 caption: add "forcing"

> We rather added "contributions" here.

P12, L294: It is still not clear what is meant with "adjustment". Further, I would suggest to rephrase the sentence or move "there" behind "show".

> Please find our statement above for clarification of the term "adjustments".

> We also suggest to leave the statement on longwave adjustments as is because it reflects our intended message.

P12, L295: At first glance -> At "a" first glance

> done!

P15, Figure 6 caption: Move "starts at 300 hPa" at the end of the sentence or write " pressure range between 300 and 10 hPa".

> done!

P18, L380: "Adjustment" to what?

> Again, it is hard for us to see what is unclear. We propose to rephrase the sentence as follows:

> *… tropospheric adjustments due to the Australian wildfire smoke are considered.*

P19, Figure 10 caption: plotted -> shown

> done!

P20,L412: low -> lower

> done!

P20, L412: make a listing instead of writing and twice? Thus, add comma after temperature and delete "and".

> done!

P20, L414: Downward impact of what? Do you mean with downward the circulation?

> We rephrased it as follows:

> *… downward influence on the troposphere …*

P20-21, L420-421: "downwelling of longwave radiation" sounds wrong.

> This is a standard term in the radiative transfer community.

P21, L427: Once again it is not clear what you mean with adjustments.

> Please find our statement above for clarification of the term "adjustments".

P22, L442: Impacts on what. Please clarify and rephrase sentence.

> done!

P24, L519: add "clouds" -> cirrus clouds

> done!

P24, L524: add "the" -> at the top

> done!

P24, L525: reported clear sky values -> where these negative? Please write clearly if negative or positive.

> We added "negative" here.

References: No consistent style used. Check ACP style and correct reference list accordingly.

> We are using the official ACP template `copernicus.bst', version 1.4 (March 2022) for rendering our bibtex entries.

> We carefully updated the journal names which had some inconsistencies.

**Response to Reviewer #2:**

The authors have taken my comments into account, and I am generally satisfied with the adjustments made to the paper. However, I do have a few additional minor suggestions:

Lines 123-125 (version with tracked changes): It should be explicitly mentioned that the perturbations in other gaseous compounds, such as ozone, carbon monoxide, etc., are not included, in addition to water vapor. Please note that the water vapor injection was detected by several spaceborne instruments.

> We updated the sentence according to your suggestion.

Line 170: You might consider adding a few references regarding the spurious impact that nudging may have on the representation of the large-scale atmospheric circulation (e.g., Davis et al., 2021 and references therein).

> Great reference! We added the following:

> *For example, in Davis et al. (2022), nudging was shown to introduce biases into the representation of residual circulation and, consequently, tracer transport.*

Figure 8: I would recommend replacing the term 'adiabatic heating' with 'advective potential temperature tendency' for clarity.

> We changed the caption of Fig. 8 as follows:

> *Similar to Fig. 7, but for the average perturbations in the tendencies of potential temperature due vertical advection which we interpret as adiabatic heating due to circulation changes.*

> However, we like to keep the term "adiabatic heating" in the figure title because it is much clearer linked to the interpretation of the underlying mathematical term and it is also much shorter than "negative tendency of potential temperature due vertical advection".

References

Davis, N. A., Callaghan, P., Simpson, I. R., and Tilmes, S.: Specified dynamics scheme impacts on wave-mean flow dynamics, convection, and tracer transport in CESM2 (WACCM6), Atmos. Chem. Phys., 22, 197–214, https://doi.org/10.5194/acp-22-197-2022, 2022